# Evaluating the transcriptional regulators of arterial gene expression via a catalogue of characterized arterial enhancers

Svanhild Nornes[1], Susann Bruche[1], Niharika Adak[1,2], Ian R McCracken[1], Sarah De Val[1,3]*

[1]Institute of Developmental and Regenerative Medicine, Department of Physiology, Anatomy and Genetics, Oxford, United Kingdom; [2]University Medical Centre Groningen, Groningen, Netherlands; [3]Ludwig Institute for Cancer Research Ltd, Nuffield Department of Medicine, University of Oxford, Oxford, United Kingdom

**\*For correspondence:**
Sarah.deval@dpag.ox.ac.uk

**Competing interest:** The authors declare that no competing interests exist.

## Abstract

The establishment and growth of the arterial endothelium require the coordinated expression of numerous genes. However, regulation of this process is not yet fully understood. Here, we combined in silico analysis with transgenic mice and zebrafish models to characterize arterial-specific enhancers associated with eight key arterial identity genes (*Acvrl1/Alk1*, *Cxcr4*, *Cxcl12*, *Efnb2*, *Gja4/Cx37*, *Gja5/Cx40*, *Nrp1*, and *Unc5b*). Next, to elucidate the regulatory pathways upstream of arterial gene transcription, we investigated the transcription factors binding each arterial enhancer compared to a similar assessment of non-arterial endothelial enhancers. These results found that binding of SOXF and ETS factors was a common occurrence at both arterial and pan-endothelial enhancers, suggesting neither are sufficient to direct arterial specificity. Conversely, FOX motifs independent of ETS motifs were over-represented at arterial enhancers. Further, MEF2 and RBPJ binding was enriched but not ubiquitous at arterial enhancers, potentially linked to specific patterns of behaviour within the arterial endothelium. Lastly, there was no shared or arterial-specific signature for WNT-associated TCF/LEF, TGFβ/BMP-associated SMAD1/5 and SMAD2/3, shear stress-associated KLF4, or venous-enriched NR2F2. This cohort of well-characterized and in vivo-verified enhancers can now provide a platform for future studies into the interaction of different transcriptional and signaling pathways with arterial gene expression.

## Editor's evaluation

This work represents a significant milestone in understanding the regulatory logic underlying arterial identity. It offers an exceptional repository of validated arterial enhancers that drive expression across different vascular compartments. Moreover, it identifies the upstream transcription factor binding sites within these arterial enhancers. As such, this study will be of broad interest to developmental biologists, vascular researchers, and cell biologists.

## Introduction

The blood vessel system consists of a highly branched network of tubes lined by endothelial cells (ECs) and hierarchically organized into arteries, veins, and capillaries. The EC layer is the first part of the blood vessel to form, initially via differentiation from mesoderm (vasculogenesis) and later from existing ECs (angiogenesis) (*Payne et al., 2024*). While the first arterial ECs arise during vasculogenesis

**eLife digest** Our blood vessels are a biological transport system that carry oxygen and nutrients to all the cells and tissues of our bodies. Each type of blood vessel has a different structure depending on its role. For example, arteries are large, strong-walled vessels that carry oxygenated blood away from the heart into the rest of the body, while veins carry blood back to the heart and lungs once all the oxygen has been used up.

All blood vessels contain an inner lining made up of cells termed endothelial cells. These cells are also important for the formation of new blood vessels, which happens via a process called angiogenesis. During angiogenesis, the endothelial lining of new vessels forms first, by 'sprouting' or 'splitting' from the endothelial cells lining existing vessels.

We know that angiogenesis is accompanied by changes in gene activity within the new endothelial cells. For example, during the development of new arteries, endothelial cells will turn on genes involved in artery formation. These changes are controlled by biological switches, which involve special proteins (called transcription factors) and DNA sequences close to specific genes (called enhancers). When the right transcription factor interacts with an enhancer for a gene, the gene 'switches on'.

Despite this, however, very few enhancers associated with arterial angiogenesis are known, and the mechanisms controlling this process are still poorly understood. Nornes et al. therefore set out to identify more arterial enhancers and study how they worked.

To identify potential enhancers, Nornes et al. first used computer-based analysis of the DNA surrounding eight genes known to be involved in artery formation. The enhancers were then tested in zebrafish and mice to confirm their ability to switch genes on in artery endothelial cells. These experiments revealed a set of 15 new arterial enhancers, which were tested in further biochemical and genetic studies to determine which transcription factors could interact with them. Several transcription factors previously thought to be involved in artery development did not appear to interact with any of the new enhancers.

This study sheds new light on the genetic control of blood vessel formation, in particular artery development. Nornes et al. hope that in the future the knowledge gained from these experiments will contribute to a better understanding of angiogenesis during early life, in health and disease.

(*Chong et al., 2011*), single-cell transcriptomics and fate-mapping experiments spanning humans, mice, and zebrafish indicate that most arterial ECs form via angiogenesis from venous and venous-like capillary ECs (*Su et al., 2018*; *Hou et al., 2022*; *McCracken et al., 2022*; *Lee et al., 2021*; *Xu et al., 2014*; *Fujita et al., 2011*; *Kaufman et al., 2015*; *Marín-Juez et al., 2016*; *McCracken et al., 2023*). During this process, a subset of ECs reduce cell-cycling and venous gene transcription, induce arterial gene expression, and migrate against flow to form into new arteries or extend existing ones. While significant alterations to gene transcription occur during this transition, the precise mix of hardwired signalling pathways and environmental stimuli regulating the differentiation of arterial ECs has been challenging to untangle and identify.

Many components of Notch signalling are selectively expressed in arterial ECs and are essential for arterial formation (e.g. the DLL4 ligand) (*Quillien et al., 2014*). However, the assumed model of Notch signalling directly activating arterial gene expression has recently been challenged by new evidence. In particular, retinal and coronary vessels lacking both endothelial MYC (a driver of metabolism and proliferation) and RBPJ (the nuclear effector of Notch signalling) can still express arterial genes and form arterial structures (*Luo et al., 2021*). This research led to a new paradigm in which Notch drives arterial EC differentiation by reducing metabolism and cell-cycle rather than by directly activating arterial genes (*Luo et al., 2021*). However, cell-cycle changes alone do not necessarily alter arterial identity, and *Myc* loss in retinal EC does not affect arterial patterning (*Wilhelm et al., 2016*). Therefore, Notch-mediated cell-cycle exit likely works alongside other regulators which directly control arterial gene transcription, while the precise role of Notch in directing arterial gene expression remains unclear.

Although numerous other regulatory pathways have been implicated in arterial transcriptional regulation, their exact contribution has been challenging to establish and none appear essential for arterial EC identity (*McCracken et al., 2023*). Both canonical WNT and TGFβ/BMP signalling pathways have

been implicated in arterialization yet ECs lacking β-catenin or SMAD4 still express arterial genes and form arterial structures (*Neal et al., 2019*; *Wythe et al., 2013*). Likewise, blood flow is required for full expression of arterial genes, yet arteries in both early embryonic and coronary vasculatures form prior to blood flow (*Su et al., 2018*; *Lee et al., 2021*; *Fang et al., 2017*; *Hwa et al., 2017*). Our knowledge of the transcription factors activating arterial gene expression is also incomplete. ETS factors are required for arterial gene activity but are also essential for vein-specific and pan-endothelial gene expression, suggesting a more general requirement for endothelial identity (*Neal et al., 2021*). The link between FOXC factors and arteries was partially predicated on binding to *Dll4* regulatory regions later found to lack arterial activity (*Wythe et al., 2013*). DACH1 potentiates arterial differentiation but is widely expressed and cannot alter EC identity (*Raftrey et al., 2021*), while arterial-enriched MECOM is linked to repression of venous gene expression rather than activation of arterial identity genes (*McCracken et al., 2022*). The evidence linking SOXF transcription factors to arterial differentiation is more extensive, with loss of either SOX17 (the SOXF factor most specific to arterial ECs) or SOX7 resulting in arterial defects (*Lilly et al., 2017*; *Kim et al., 2016*; *Corada et al., 2013*; *Zhou et al., 2015*). Whilst losing a single SOXF factor does not entirely compromise the arterial programme, arterial differentiation appears absent after compound *Sox17;Sox18* and *Sox7;Sox17;Sox18* deletion, although this occurs alongside significantly impaired angiogenesis and severe vascular hyperplasia (*Lilly et al., 2017*; *Kim et al., 2016*; *Corada et al., 2013*; *Zhou et al., 2015*). Additionally, the manner in which SOXF factors contribute to the specific activation of arterial genes is still unknown: while SOX17 is considered arterial-specific by late fetogenesis, both SOX7 and SOX18 are more widely expressed, and all SOXF factors bind the same motifs (*Francois et al., 2010*). It is also unclear whether SOXF factors primarily act upstream of Notch signalling (and subsequent cell-cycle-related control of arterial differentiation), or whether they more widely activate arterial gene expression. Direct SOXF binding is best characterized at Notch pathway enhancers *Dll4in3*, *Dll4-12,* and *Notch1+16,* and the arterial defects seen after *Sox17* deletion were attributed to a requirement for SOX17 in activation of *Notch1* and *Dll4* (*Payne et al., 2024*; *Corada et al., 2013*; *Sacilotto et al., 2013*). However, SOXF motifs are also required for the activity of the arterial-specific *ECE1in1* enhancer and are associated with coronary arterial *Nestin* expression (*González-Hernández et al., 2020*; *Robinson et al., 2014*).

In this article, we identify a cohort of arterial enhancers associated with eight key arterial identity genes, combining in silico analysis with verification and characterization in transgenic models. We then use sequence analysis and DNA-protein binding surveys to investigate the involvement of many endothelial- and arterial-associated transcription factors in arterial enhancer binding, and to compare this pattern with that seen at pan-endothelial and venous enhancers. Our results indicate that ETS and SOXF factors play a general role in endothelial gene transcription, suggest a role for FOX factors more selectively in arterial activation, and link both RBPJ and MEF2 factors to a limited number of arterial genes, potentially related to specific expression patterns. This cohort of well-characterized, in vivo-verified enhancers can also now be used as a platform for future studies into the interaction of different transcriptional and signalling pathways with specific arterial genes and with subtype-specific gene expression within the endothelium more generally. Additionally, our data provides a useful training set for attempts to more accurately classify endothelial enhancers genome-wide.

## Results

### In silico identification of putative enhancers for key arterial identity genes

Transcription factors primarily regulate endothelial gene transcription through binding to enhancers (*cis*-regulatory elements) (*Payne et al., 2024*). Consequently, analysis of enhancer sequences can elucidate the precise combination of transcription factors, and cognate upstream signalling pathways, involved in different patterns of gene expression. One of the main challenges in understanding arterial regulation has been a paucity of characterized enhancers for key arterial genes. For example, of the 16 genes used to define mouse coronary arterial EC identity in single-cell transcriptomics (*Raftrey et al., 2021*), only four have in vivo-verified enhancers (*Dll4*, *Hey1*, *Notch1*, and *Acvrl1*). Three of these are genes in the Notch pathway and are either self-regulated by Notch/RBPJ (*Dll4-12* and *Hey1-18*) (*Sacilotto et al., 2013*; *Watanabe et al., 2020*) or lack specificity during early coronary arterial specification (*Notch1+16*) (*Payne et al., 2019*). The fourth enhancer, for *Acvrl1*, is of a size (9 kb) that precludes

analysis (*Seki et al., 2004*). Beyond this, there are only four other in vivo-validated arterial enhancers described in the literature, for *Ece1, Flk1, Sema6d,* and *Sox7*. Of these, only the *Ece1in1* and *Flk1in10* arterial enhancers, both associated with genes not specific to arterial ECs, have been analysed at the level of transcription factor binding (*Robinson et al., 2014*; *Zhou et al., 2017*). It is therefore clear that a better understanding of the regulatory pathways directing arterial differentiation requires the identification and characterization of a larger number of arterial enhancers orchestrating the expression of key arterial identity genes. To identify a cohort of such enhancers, we looked in the loci of eight non-Notch genes: *Acvrl1*(ALK1) *Cxcr4, Cxcl12, Efnb2, Gja4*(CX37), *Gja5* (CX40), *Nrp1,* and *Unc5b*. Although not a definitive list of arterial identity genes, single-cell transcriptomic analysis indicates these genes are all significantly enriched in arterial ECs (*Hou et al., 2022*; *Raftrey et al., 2021*), and they are commonly used to define arterial EC populations in mouse and human scRNAseq analysis (*Hou et al., 2022*; *McCracken et al., 2022*; *Raftrey et al., 2021*; *Phansalkar et al., 2021*). Additionally, the genes selected here are also equally split between the two arterial subgroups identified by single-cell transcriptomics: *Cxcr4*, *Efnb2*, *Gja4,* and *Unc5b* included in the earlier expressed arterial plexus/pre-arterial EC subgroup, *Acvrl1*, *Cxcl12*, *Gja5*, and *Nrp1* restricted to the mature arterial EC subgroup (*Hou et al., 2022*; *Raftrey et al., 2021*). We did not exclude genes implicated in angiogenesis/expressed in sprouting ECs as these overlapped with genes within the pre-arterial EC subgroup.

To identify putative enhancers in silico, we used five published datasets detailing different enhancer-associated chromatin marks: (i) open chromatin as assessed by ATAC-seq in primary mouse adult aortic ECs (MAECs) from *Engelbrecht et al., 2020*; (ii) open chromatin as assessed by ATAC-seq in mouse postnatal day 6 (P6) retina ECs (MRECs) from *Yanagida et al., 2020*; (iii) enriched EP300 binding in Tie2Cre+ve cells from embryonic day (E) 11.5 mouse embryos from *Zhou et al., 2017*; (iv) enriched H3K27Ac and/or H3K4Me1 in human umbilical vein ECs (HUVECs, data available on the UCSC Genome Browser; *Rosenbloom et al., 2013*); and (v) open chromatin regions assessed by DNAseI hypersensitivity in HUVECs and dermal-derived neonatal and adult blood microvascular ECs (HMVEC-dBl-neo/ad) comparative to non-ECs (UCSC Genome Browser; *Rosenbloom et al., 2013*; *Figure 1*). A retrospective analysis of 32 previously described mammalian in vivo-validated EC enhancers (*Payne et al., 2024*), which included eight arterial enhancers, found that 31/32 were marked by at least one enhancer mark in both human and mouse samples (including 8/8 of arterial enhancers) (see *Table 1*). We analysed the loci of our target arterial genes to identify putative enhancers using these enhancer marks. For arterial genes robustly transcribed in both human and mouse EC datasets (determined by open chromatin/H3K4Me3 at the promoter region), we defined a putative enhancer as a region containing at least one enhancer mark in both mouse and human ECs. Because *Cxcr4*, *Cxcl12*, and *Gja5* were poorly transcribed in the human cell lines studied here, for these genes the putative enhancer definition was relaxed to include regions containing two enhancer marks in mouse ECs with no marks in human cells. Orthologous human enhancer sequences were identified for every enhancer using the Vertebrate Multiz Alignment & Conservation Track on the UCSC Genome Browser. Each putative enhancer was named according to their neighbouring arterial gene and distance from the transcriptional start site (TSS) in mice (e.g. the putative *Efnb2-112* enhancer is 112 kb upstream of the mouse *Efnb2* TSS). In total, this analysis considered over 110 regions and identified 41 putative enhancers for further testing (*Figure 1*, *Table 2—source data 1*). We also assessed seven regions previously identified as potential enhancers for *Efnb2*, *Nrp1,* and *Cxcr4* but whose independent activity was never validated in vivo (*Grego-Bessa et al., 2007*; *Yamamizu et al., 2010*; *Tsaryk et al., 2022*; *Stewen et al., 2024*). None of these met our putative enhancer threshold for further testing: two regions were associated with no enhancer marks, three had a single enhancer mark in HUVECs, one had non-specific enhancer marks in human cells only (*Nrp1+76/NRP1A; Yamamizu et al., 2010*) and one contained enhancer marks in human ECs only (*Cxcr4-117/CXCR4+125; Tsaryk et al., 2022*; *Figure 1*, *Table 2—source data 1*).

Out of the 41 putative enhancers identified here, 3 had been previously investigated in vivo: *Nrp1+28*, which was able to drive robust pan-endothelial expression of the *lacZ* reporter gene in transgenic mouse embryos (*De Val et al., 2008*); and *Acvrl-5* and *Acvrl1-1/p*, which were silent in a similar mouse assay (*Seki et al., 2004*; *Table 2*). Additionally, *Acvrl1+6* is contained within a nine kilobase region known to direct arterial-specific expression of *lacZ* in transgenic mice (*Seki et al., 2004*). However, because of the size of this original piece, we treated the smaller *Acvrl1+6* enhancer as untested.

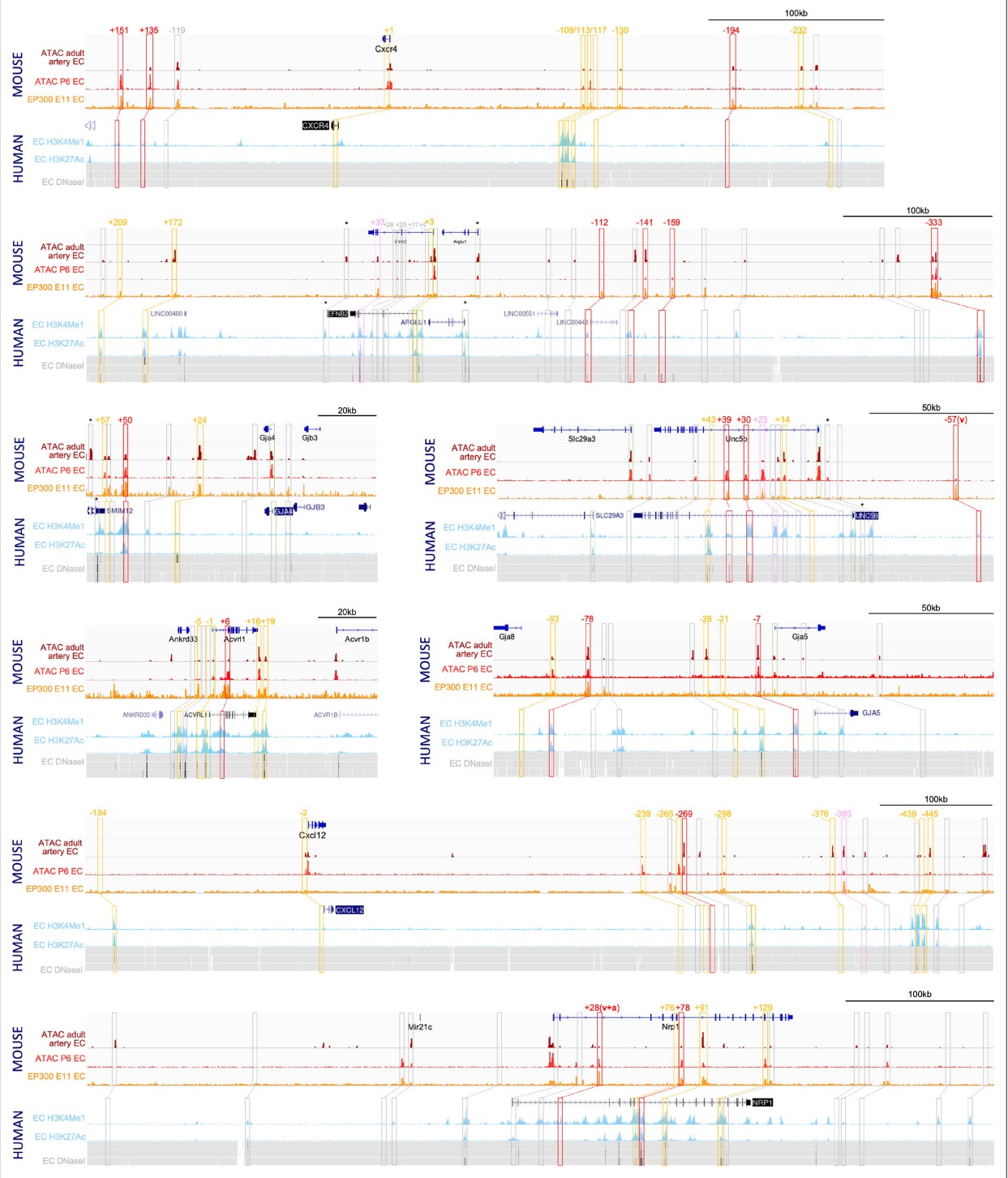

**Figure 1.** Analysis of enhancer marks around eight arterial-expressed genes identifies putative arterial enhancers. Enhancer marks from mouse tissue include: dark red 'ATAC adult artery EC' denotes open chromatin assessed by ATAC-seq in primary adult aortic endothelial cells (ECs) *Engelbrecht et al., 2020*; bright red 'ATAC P6 EC' denotes open chromatin assessed by ATAC-seq in postnatal day 6 retinal ECs *Yanagida et al., 2020*; orange 'EP300 E11 EC' denotes enriched EP300 binding in Tie2Cre+ve cells in embryonic day 11.5 embryos (*Zhou et al., 2017*). Enhancer marks from human cells include: light blue peaks denotes enriched H3K27Ac and H3K4Me1 in human umbilical vein ECs (HUVECs) (UCSC Genome Browser; *Rosenbloom et al., 2013*); grey heat map denotes open chromatin regions assessed by DNAseI hypersensitivity in HUVECs (upper line) and dermal-derived neonatal and adult blood microvascular ECs (HMVEC-dBl-neo/ad, middle and bottom line) (UCSC Genome Browser). Red, pink, and orange solid boxes indicate regions fitting putative enhancers criteria and selected for analysis (red/pink/orange indicates strong/weak/silent EC activity in transgenic models, see *Table 2* and *Figures 2 and 3*). Numbers represent approximate distance from TSS. Orange dashed boxes indicates regions below the putative

*Figure 1 continued on next page*

*Figure 1 continued*

enhancer threshold but included in transgenic assays as controls, grey boxes indicate regions below the putative enhancer threshold and not tested. * indicates that enhancer marks were not specific for ECs but rather found in many cell types.

The online version of this article includes the following figure supplement(s) for figure 1:

**Figure supplement 1.** The mouse Dll4-12 enhancer directs arterial expression of reporter genes in the vasculature of transgenic zebrafish (**A**) and mice (**B**).

**Figure supplement 2.** Analysis of putative enhancers in F0 mosaic Tol2 transgenic zebrafish identifies 15 enhancers able to drive robust GFP activity in arterial endothelial cells (ECs) (other weaker/ non-arterial enhancers are detailed in *Figure 1—figure supplement 3*).

**Figure supplement 3.** Additional images of transgenic fish expressing the Unc5b-57:GFP, Cxcl12+383:GFP, Efnb2-37, Unc5b+23, Cxcl12+117, Efnb2-112 and Efnb2-159 transgenes.

**Figure supplement 4.** Enhancer marks in cultured human arterial endothelial cells (ECs) around the eight target arterial gene loci.

## Transgenesis in zebrafish confirms arterial activity for a subset of putative enhancers

It is well established that DNA sequences associated with enhancer marks do not necessarily act as enhancers (e.g. by independently activating gene transcription). To establish the ability of the 38 untested putative enhancers to drive arterial EC activity, we first analysed the activity of each in F0 Tol2-mediated mosaic transgenic zebrafish embryos (*Kawakami, 2005*). Similar zebrafish assays have been conducted with five of the eight previously identified arterial enhancers, and in each case these mammalian enhancer sequences were able to drive GFP expression in arterial ECs in zebrafish embryos (*Figure 1*, *Figure 1—figure supplement 1*; *Sacilotto et al., 2013*; *Chiang et al., 2017*; *Becker et al., 2016*; *Andersson et al., 2014*). Here, the mouse sequence of each of the 38 putative arterial enhancers was cloned upstream of the E1b minimal promoter and GFP reporter gene and used to generate mosaic transgenic embryos, which were examined at 2 days post fertilization (dpf) (*Kawakami, 2005*). This analysis identified 19 enhancers able to drive GFP expression in ECs of transgenic zebrafish, defined as vascular GFP expression in more than 5% of injected embryos (*Table 2*). Sixteen of these active enhancers were able to drive robust and reproducible patterns of endothelial GFP expression in F0 mosaic transgenics (*Figure 1—figure supplement 2*) from which we were able to establish stable transgenic zebrafish lines. Of these, 15/16 enhancer:GFP transgenes were clearly expressed in arteries and were therefore selected for further analysis (*Figure 2* and *Figure 3*, genome coordinates *Table 2—source data 2*). This included at least one enhancer for each of our eight target genes. The outlier, *Unc5b-57*, was active only in the anterior portion of the caudal vein and was excluded from further analysis (*Figure 1—figure supplement 3A–B*). The remaining three enhancers were able to drive only weak GFP and/or were limited to a small number of ECs in all analysed zebrafish (*Table 2*, *Figure 1—figure supplement 3C*). GFP expression pattern within the vasculature could not be determined and no stable lines could be established; therefore, these weak vascular enhancers were not followed up further. Another 19 putative enhancer regions did not consistently drive detectable GFP expression in ECs (*Table 2*). These were designated inactive regions without developmental enhancer activity.

In addition to the 38 putative enhancers identified through our screen, we tested another 11 regions that fell below our enhancer threshold. This included regions with enhancer marks in mice only or human only, regions with only one enhancer mark, and the two regions previously implicated as enhancers (*Nrp1+76* and *Cxcr4-117*; *Tsaryk et al., 2022*). Of these 11 regions, only *Cxcr4-117* was able to drive any detectable expression in ECs. However, this was seen in only 9 of 209 injected embryos (<5%) and was limited to 1–2 ECs in each F0 fish (*Table 2*, *Figure 1—figure supplement 3D*). A human version of this enhancer, *CXCR4-125* (identified as an enhancer in *Tsaryk et al., 2022*), was also tested but was not able to drive any detectable GFP expression (*Table 2*), and this enhancer region was therefore not further analysed.

## Assessing arterial enhancer activity patterns in transgenic models

We next examined the activity of our 15 strong arterial enhancers in further detail. To determine the expression pattern of these enhancers in stable transgenic zebrafish lines, we first assessed GFP expression in the trunk vasculature at 2–3 dpf (*Figures 2–4*). Interestingly, whilst all 15 enhancers were preferentially active in trunk arteries, the pattern of GFP activity within the arterial tree varied.

**Table 1.** Enhancer marks around 32 known in vivo-characterized endothelial enhancers (all described in *Payne et al., 2024*). Red text indicates arterial enhancers.

| Enhancer | hg19 coordinates | H DNAseI | H histone | Mm9 coordinates | M artery ATAC | M retina ATAC | M E11 p300 |
|---|---|---|---|---|---|---|---|
| *Apln+28* | chrX:128,756,756–128,757,160 | Yes | Yes | chrX:45,359,306–45,359,632 | No | Yes | Yes |
| Dab2-240 | chr5:39,755,997–39,756,596 | Yes* | Yes | chr15:6,009,719–6,010,138 | No | No | Yes |
| *Dll4in3* | chr15:41,222,881–41,223,570 | Yes | Yes | chr2:119,152,838–119,153,684 | Yes | Yes | Yes |
| *Dll4-12* | chr15:41,210,706–41,211,825 | Yes | Yes | chr2:119,140,274–119,141,353 | Yes | Yes | Yes |
| *Ece1in1* | chr1:21,606,038–21,607,057 | Yes | Yes | chr4:137,475,719–137,476,738 | Yes | Yes | Yes |
| Egfl7-9 | chr9:139,540,750–139,541,299 | Yes | Yes | chr2:26,427,513–26,427,707 | Yes | Yes | Yes |
| Egfl7-2 | chr9:139,550,292–139,550,891 | Yes | Yes | chr2:26,434,087–26,434,301 | Yes | Yes | Yes |
| Emcn-22 | chr4:101,460,885–101,461,224 | No | Yes | chr3:136,984,547–136,984,951 | Yes | No | No |
| Eng-8 | chr9:130,624,538–130,624,804 | Yes | Yes | chr2:32,493,606–32,493,823 | Yes | Yes | Yes |
| Eng +9 | chr9:130,607,199–130,607,657 | Yes | Yes | chr2:32,511,282–32,511,641 | Yes | Yes | Yes |
| Ephb4-2 | chr7:100,426,337–100,427,259 | Yes | Yes | chr5:137,789,910–137,790,581 | No | No | Yes |
| Fli1+12 | chr11:128,575,436–128,575,782 | Yes | Yes | chr9:32,337,295–32,337,538 | Yes | Yes | Yes |
| Flk1+3 | chr4:55,987,345–55,987,920 | Yes | Yes | chr5:76,370,627–76,371,056 | No | No | Yes |
| *Flk1in10* | chr4:55,972,978–55,973,903 | Yes | No | chr5:76,357,891–76,358,715 | No | Yes | Yes |
| Flt4+26 | chr5:180,050,291–180,050,684 | No | Yes | chr11:49,445,777–49,446,175 | Yes | Yes | Yes |
| Foxp1+138 | chr3:71,493,515–71,493,886 | Yes | Yes | chr6:99,338,958–99,339,515 | No | Yes | Yes |
| Gata2+9 | chr3:128,201,971–128,202,273 | Yes | Yes | chr6:88,153,077–88,153,386 | Yes | Yes | Yes |
| *Hey1-18* | chr8:80,695,610–80,697,109 | Yes | Yes | chr3:8,685,099–8,685,821 | Yes | Yes | Yes |
| Hlx-3 | chr1:221,049,978–221,050,354 | Yes | Yes | chr1:186,558,918–186,559,303 | Yes | Yes | Yes |
| Mef2F10 | chr5:88,110,980–88,111,253 | Yes | Yes | chr13:83,721,761–83,722,057 | No | Yes | Yes |
| Mef2F7 | chr5:88,123,031–88,123,357 | Yes | Yes | chr13:83,711,180–83,711,509 | Yes | Yes | Yes |
| *Notch1+16* | chr9:139,424,543–139,424,953 | Yes | Yes | chr2:26,346,100–26,346,671 | Yes | Yes | Yes |
| Notch1+33 | chr9:139,406,356–139,406,655 | Yes | Yes | chr2:26,330,559–26,330,785 | Yes | Yes | Yes |
| CoupTFII-965 | chr15:95,908,708–95,909,240 | Yes | Yes | chr7:78,456,407–78,456,767 | Yes | Yes | Yes |

*Table 1 continued on next page*

Table 1 continued

| Enhancer | hg19 coordinates | H DNAseI | H histone | Mm9 coordinates | M artery ATAC | M retina ATAC | M E11 p300 |
|---|---|---|---|---|---|---|---|
| Nrp1+28 | chr10:33,590,960–33,591,499 | No | Yes | chr8:130,911,132–130,911,551 | Yes | Yes | Yes |
| Nrp2+26 | chr2:206,573,202–206,573,523 | Yes | Yes | chr1:62,776,231–62,776,553 | Yes | Yes | Yes |
| Pdgfrb +18 | chr5:149,516,883–149,517,356 | No | Yes | chr18:61,219,244–61,219,566 | No | No | No |
| Epcr-5 | chr20:33,754,176–33,754,585 | Yes | Yes | chr2:155,568,588–155,569,127 | Yes | No | Yes |
| Sema6d-55 | chr15:47,958,023–47,958,764 | Yes | Yes | chr2:124,380,522–124,381,285 | Yes | Yes | Yes |
| Sox7+14 | chr8:10,573,085–10,574,291 | Yes | Yes | chr14:64,576,271–64,577,533 | Yes | Yes | Yes |
| Tal +19 | chr1:47,677,539–47,677,958 | Yes | Yes | chr4:114,748,131–114,748,530 | Yes | Yes | Yes |
| Tal1-4 | chr1:47,701,050–47,701,347 | No | Yes | chr4:114,725,243–114,725,552 | Yes | Yes | Yes |

* "H DNAseI" indicates open chromatin regions as defined by DNAseI hypersensitivity in HUVECs, HMVEC-dBl-neo and HMVEC-dBl-ad comparative to non-ECs and relative to surrounding region (UCSC genome browser[38]); "H histone" indicates relatively enriched binding of H3K27Ac and/or H3K4Me1 in HUVECs (UCSC genome browser[38]), * indicates this extends to many non-EC lines as well; "M artery ATAC" indicates regions of relatively open chromatin assessed by ATAC-seq in primary mouse adult aortic ECs (MAECs) (Engelbrecht et al[35]); "M retina ATAC" indicates regions of relatively open chromatin assessed by ATAC-seq in mouse postnatal day 6 (P6) retina ECs (MRECs) (Yanagida et al[36]); "M E11 p300" indicates regions relatively enriched for EP300 binding in Tie2Cre+ve cells from embryonic day (E)11.5 mouse embryos (Zhou et al[37]). * indicates that enhancer marks were not specific for ECs but rather found in many cell types.'M artery ATAC' indicates regions of relatively open chromatin assessed by ATAC-seq in primary mouse adult aortic ECs (MAECs) **Engelbrecht et al., 2020**; 'M retina ATAC' indicates regions of relatively open chromatin assessed by ATAC-seq in mouse postnatal day 6 (P6) retina ECs (MRECs) **Yanagida et al., 2020**; 'M E11 p300' indicates regions relatively enriched for EP300 binding in Tie2Cre+ve cells from embryonic day (E) 11.5 mouse embryos (**Zhou et al., 2017**). *indicates that enhancer marks were not specific for ECs but rather found in many cell types.

By 3 dpf, five enhancers were broadly active in both the dorsal aorta and the intersegmental arteries sprouting off the aorta, whilst four enhancers were restricted to the dorsal aorta only, and five were restricted to the intersegmental arteries only. In addition, we saw some differences in arterial specificity. By 3 dpf, 6/9 enhancers active in the dorsal aorta showed no expression in the cardinal vein, whilst the other three were also weakly expressed in anterior or posterior segments of the cardinal vein (*Figures 3 and 4*). Transgene expression in the intersegmental vessels was more complicated to interpret as these vessels form by angiogenic sprouting from the dorsal aorta and many arterial genes are also expressed during angiogenic sprouting (including *Efnb2*, *Cxcr4*, *Dll4*, *Nrp1*, and *Sox7*). Consequently, GFP expression seen during early intersegmental sprouting could potentially reflect angiogenic or arterial expression (*Sacilotto et al., 2013*; *Yamamizu et al., 2010*). However, our analysis demonstrated that all enhancers active in the intersegmental vessels domain became either arterial-enriched or arterial-specific by 3 dpf: all 12 enhancers active in the intersegmental vessels were strongly enriched in the 3 dpf intersegmental arteries (as defined by direct connections to the dorsal aorta rather than the cardinal vein), with only 4/12 showing weaker expression in some intersegmental veins (*Figures 2–4*). Overall, this analysis suggests that, in the zebrafish trunk vasculature, 10/15 of our enhancers were arterial-specific whilst 5/15 were arterial-enriched.

Arterial development in the zebrafish trunk at the timepoints investigated primarily occurs via vasculogenesis and arterial-to-arterial sprouting, as opposed to the vein/capillary origin of many mammalian arterial ECs (*Su et al., 2018*; *Hou et al., 2022*; *Red-Horse and Siekmann, 2019*). In order to investigate whether our enhancers were also active in arterial vessels formed directly from venous ECs, we also looked at expression in the developing zebrafish eye. In the zebrafish ocular system, venous ECs from the dorsal ciliary vein (DCV) sprout to form the nasal ciliary artery (*Kaufman et al., 2015*; *Hasan et al., 2017*; *Hashiura et al., 2017*). Analysis of our enhancers in the 2–3 dpf eye found that 11/15 were active in the nasal ciliary artery, of which only two were also expressed in the dorsal ciliary vein (*Figure 3—figure supplement 1*, *Figure 4*). Only four enhancers active in the trunk arteries

**Table 2.** Summary of putative enhancer activity in mosaic Tol2 transgenic zebrafish.
'In vivo classification' indicates the results of this screen, 'in silico classification' indicates designation from dataset from *Sissaoui et al., 2020*, as defined by relative enhancer and promoter marks in HUVECs vs. HUAECs.

| Enhancer | # Injected/# any EC GFP | In vivo classification | In silico classification |
|---|---|---|---|
| *Cxcr4-232* | 163/0 | Inactive | *Uncalled* |
| **Cxcr4-194** | 46/25 | Arterial enhancer | *Uncalled* |
| *Cxcr4-130* | 95/0 | Inactive | *Uncalled* |
| *Cxcr4-117^* | 209/9* | Inactive | *Uncalled* |
| *hCxcr4-117^/CXCR4-125* | 81/0 | Inactive | *Uncalled* |
| *Cxcr4-113* | 300/0 | Inactive | *Common EC enhancer* |
| *Cxcr4-109* | 89/0 | Inactive | *Common EC enhancer* |
| *Cxcr4+1* | 33/0 | Inactive | *Uncalled* |
| *Cxcr4+119* | 187/0 | Inactive | *Arterial enhancer* |
| **Cxcr4+135** | 69/56 | Arterial enhancer | *Uncalled* |
| **Cxcr4+151** | 96/50 | Arterial enhancer | *Arterial TSS* |
| **Efnb2-333** | 152/93 | Arterial enhancer | *Common EC enhancer* |
| **Efnb2-159** | 247/217 | Arterial enhancer | *Common EC enhancer* |
| **Efnb2-141** | 74/36 | Arterial enhancer | *Arterial TSS* |
| **Efnb2-112** | 65/30 | Arterial enhancer | *Arterial TSS* |
| *Efnb2+3* | 92/0 | Inactive | *Common EC TSS* |
| **Efnb2+37** | 114/18* | Weak enhancer* | *Common EC enhancer* |
| *Efnb2+172* | 63/0 | Inactive | *Common EC enhancer* |
| *Efnb2+209* | 158/0 | Inactive | *Common EC enhancer* |
| *Gja4+24* | 52/0 | Inactive | *Uncalled* |
| **Gja4+50** | 232/187 | Arterial enhancer | *Common EC enhancer* |
| *Gja4+57* | 192/0 | Inactive | *Uncalled* |
| **Unc5b-57** | 50/21 | Venous enhancer | *Uncalled* |
| *Unc5b+14* | 61/0 | Inactive | *Uncalled* |
| **Unc5b+23** | 82/16* | Weak enhancer* | *Arterial enhancer* |
| **Unc5b+30** | 96/79 | Arterial enhancer | *Arterial enhancer* |
| **Unc5b+39** | 96/56 | Arterial enhancer | *Arterial enhancer* |
| *Unc5b+43* | 111/0 | Inactive | *Arterial enhancer* |
| *Acvrl1-5* | ***Seki et al., 2004*** | Inactive | *Common EC enhancer* |
| *Acvrl1-1/p* | ***Seki et al., 2004*** | Inactive | *Common EC TSS* |
| **Acvrl1+6** | 127/49 | Arterial enhancer | *Common EC TSS* |
| *Acvrl1+16* | 205/0 | Inactive | *Common EC TSS* |
| *Acvrl1+19* | 95/0 | Inactive | *Common EC TSS* |
| *Cxcl12-184* | 64/0 | Inactive | *Common EC enhancer* |
| *Cxcl12-2* | 32/0 | Inactive | *Common EC enhancer* |
| *Cxcl12+239* | 70/1 | Inactive | *Uncalled* |

*Table 2 continued on next page*

*Table 2 continued*

| Enhancer | # Injected/# any EC GFP | In vivo classification | In silico classification |
|---|---|---|---|
| *Cxcl12+265* | 42/1 | Inactive | *Uncalled* |
| *Cxcl12+269* | 163/63 | Arterial enhancer | *Uncalled* |
| *Cxcl12+298* | 51/0 | Inactive | *Common EC enhancer* |
| *Cxcl12+376* | 149/0 | Inactive | *Uncalled* |
| *Cxcl12+383* | 152/37* | Weak enhancer* | *Uncalled* |
| *Cxcl12+439* | 73/0 | Inactive | *Common EC enhancer* |
| *Cxcl12+445* | 145/0 | Inactive | *Common EC TSS* |
| *Gja5-7* | 66/39 | Arterial enhancer | *Arterial enhancer* |
| *Gja5-21*[†] | 38/0 | Inactive | *Common EC enhancer* |
| *Gja5-28* | 156/0 | Inactive | *Uncalled* |
| *Gja5-78* | 76/62 | Arterial enhancer | *Common EC enhancer* |
| *Gja5-93* | 253/0 | Inactive | *Arterial enhancer* |
| *Nrp1+28* | *De Val et al., 2008* | Pan-EC enhancer | *Common EC enhancer* |
| *Nrp1+76* | 54/0 | Inactive | *Common EC enhancer* |
| *Nrp1+78* | 191/34 | Arterial enhancer | *Common EC enhancer* |
| *Nrp1+91* | 109/0 | Inactive | *Common EC enhancer* |
| *Nrp1+129* | 110/0 | Inactive | *Common EC enhancer* |
| *DLL4-12* | *Sacilotto et al., 2013* | Arterial enhancer | *Common EC enhancer* |
| *DLL4in3* | *Sacilotto et al., 2013* | Arterial enhancer | *Common EC enhancer* |
| *ECE1in1* | *Robinson et al., 2014* | Arterial enhancer | *Common EC enhancer* |
| *Flk1in10* | *Becker et al., 2016* | Arterial enhancer | *Uncalled* |
| *Hey1-18* | *Watanabe et al., 2020* | Arterial enhancer | *Common EC enhancer* |
| *NOTCH1+16* | *Chiang et al., 2017* | Arterial enhancer | *Common EC enhancer* |
| *Sema6d-55* | *Zhou et al., 2017* | Arterial enhancer | *Common EC enhancer* |
| *SOX7+14* | *Zhou et al., 2017* | Arterial enhancer | *Common EC enhancer* |

*indicates only limited expression in a very small number of ECs. *Acvrl1-5* and *Acvrl1-1/p* were previously investigated by **Seki et al., 2003**, *Nrp1+28* by **De Val et al., 2008**, *Notch1+16* by **Chiang et al., 2017**, *Dll4-12* and *Dll4in3* by **Sacilotto et al., 2013**, *Flk1in10* by **Becker et al., 2016**, *Sema6d-55* by **Zhou et al., 2017**, *Sox7+14* by **Zhou et al., 2017**; **Andersson et al., 2014**, *Ece1in10* by **Robinson et al., 2014**, and *Hey1-18* by **Watanabe et al., 2020**.

[†]After this analysis but prior to this publication, *Gja5-21* was shown to direct expression in the zebrafish endocardium at 4 dpf (**Chiang et al., 2023**).

The online version of this article includes the following source data for table 2:

**Source data 1.** Enhancer marks in different human and mouse ECs at putative enhancer regions within the loci of eight arterial genes.

**Source data 2.** Genome locations for all enhancer (and human orthologues) investigated in this paper.

showed no ocular expression. This did not correspond to any particular arterial expression pattern in the trunk vasculature, suggesting that the two behaviours may not be transcriptionally linked.

We also investigated arterial enhancer activity in adult zebrafish. For this, we examined uninjured adult dorsal fins from 13 enhancer:GFP adult stable lines, alongside two previously studied arterial enhancers (Dll4in3 and Dll4-12) and the pan-EC marker line *tg(fli1a:GFP)* previously used for similar analysis (*Xu et al., 2014*; *Figures 4 and 5*). 11/13 novel arterial enhancers and 2/2 known arterial enhancers were active in the adult fin arteries and/or arterial sprouts (*Figure 5*). Although no adult F1 fish were available for the *Ephb2-112:GFP* and *Efnb2-159:GFP* transgenes, analysis in 4-week-old juveniles also confirmed arterial fin expression (*Figure 1—figure supplement 3E*). Consequently, it is

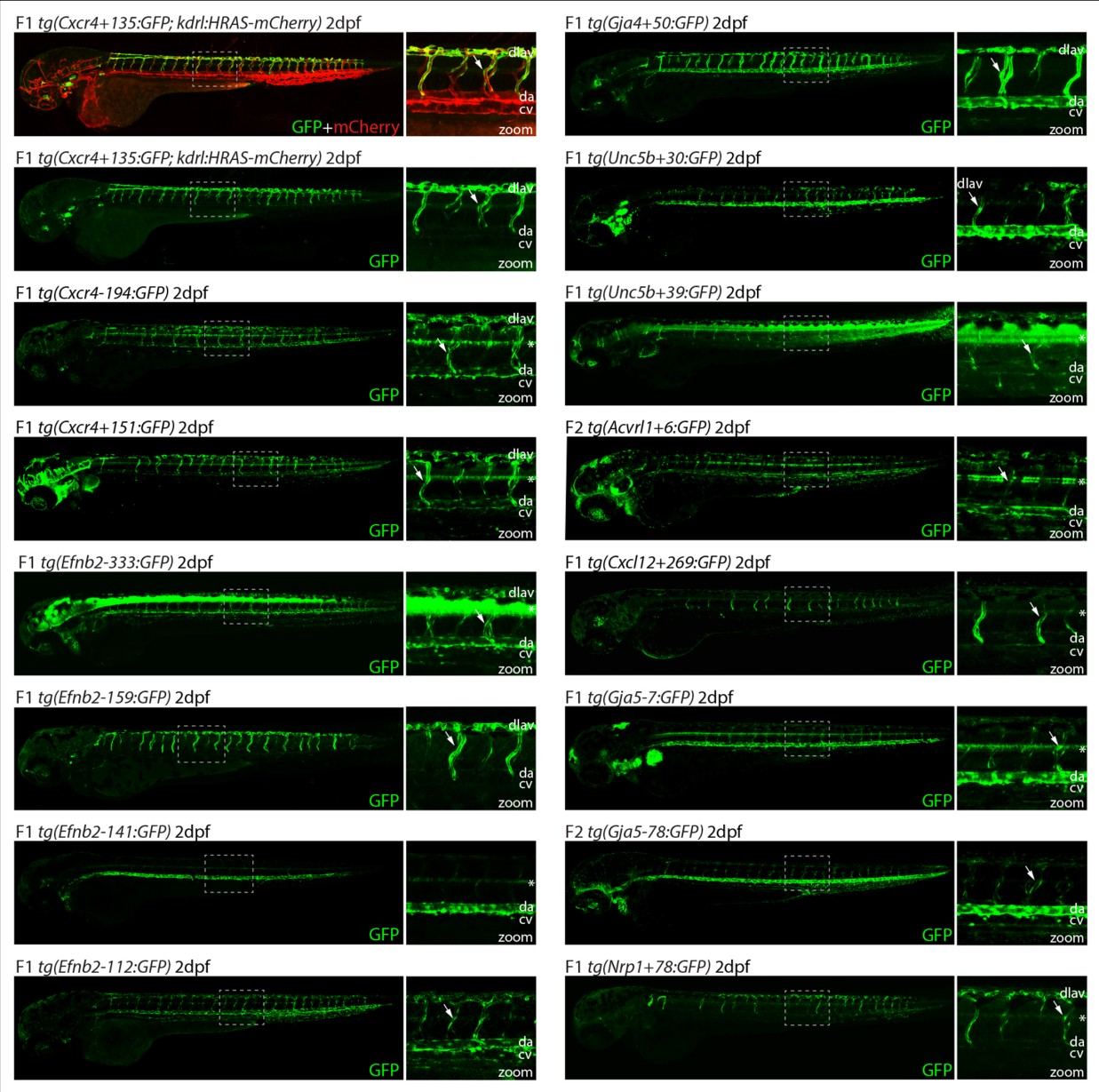

**Figure 2.** Analysis in 2 dpf transgenic zebrafish indicates arterial activity of fifteen novel enhancer:GFP transgenes. Grey dashed box indicates region of zoom, da indicates dorsal aorta, cv indicates cardinal vein, dlav indicates dorsal longitudinal anastomotic vessel, white arrow indicates intersegmental vessels, and * indicates expression in neural tube. *tg(Cxcr4-135:GFP)* was crossed with *tg(kdrl:HRAS-mCherry),* which expresses mCherry in all blood vascular ECs and is shown here on the top line as a guide to vessel structure at this timepoint. F1/2 indicates generation of embryo.

clear that our developmentally active arterial enhancers are largely also able to direct arterial patterns of expression in the adult, more quiescent, vasculature.

Lastly, we investigated whether these enhancers (all mouse sequences) also directed arterial activity in transgenic mice, selecting five enhancers active in zebrafish for further analysis. In each case, the enhancer was able to drive activity of the *lacZ* reporter gene in arterial ECs in E14.5 F0 transgenic mouse embryos in patterns similar to the previously described arterial *Dll4in3:lacZ* transgene (*Figure 6*). We additionally tested one enhancer which was only weakly active in transgenic zebrafish (*Efnb2-37*). No endothelial activity was seen for this enhancer in transgenic mice (*Figure 6—figure supplement 1*). In combination, this transgenic zebrafish and mouse analysis indicates that we have successfully identified a cohort of enhancers directing gene expression to arterial ECs, accurately reflecting the expression of their cognate genes in mammals.

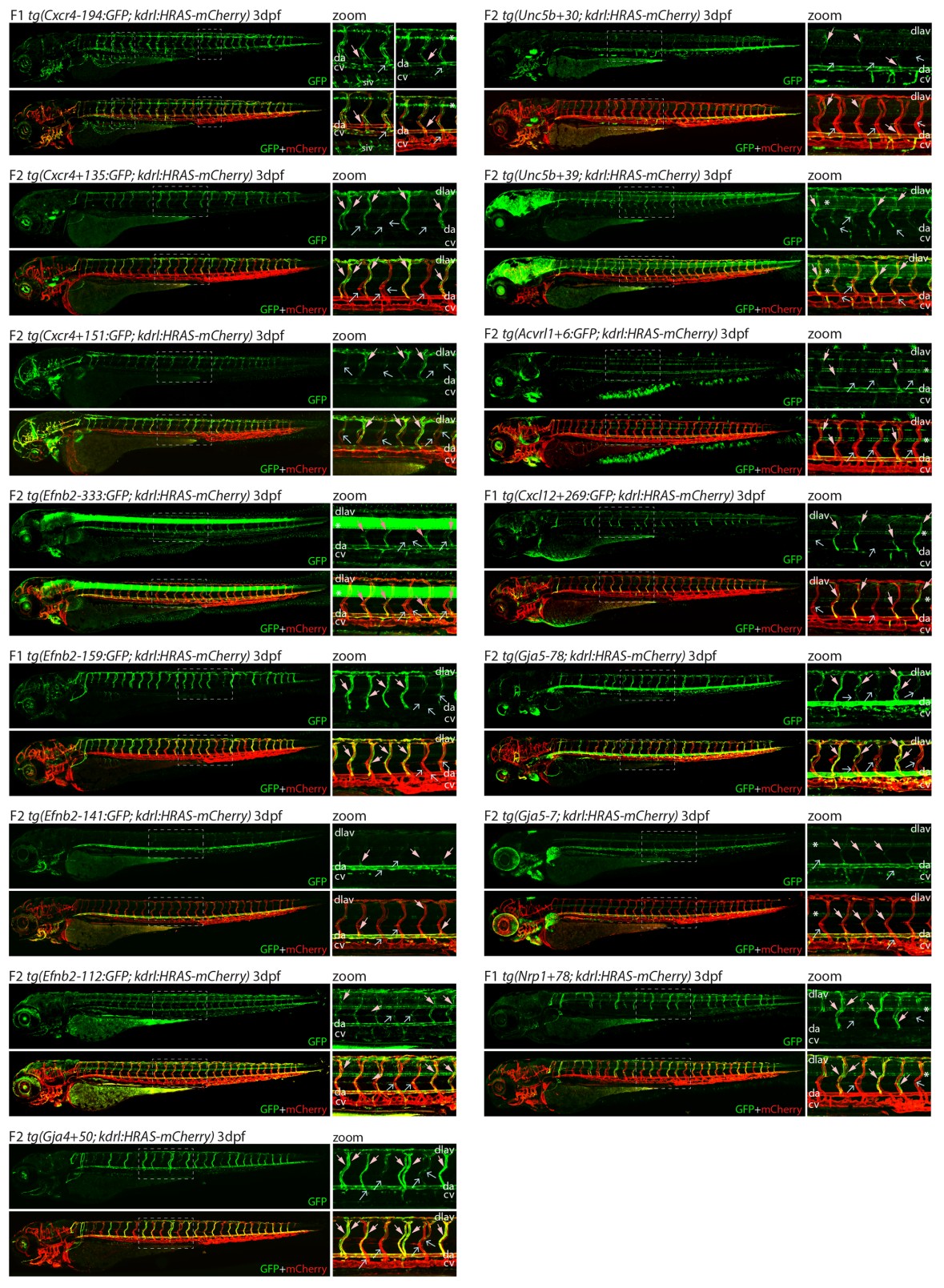

**Figure 3.** Analysis in 3 dpf transgenic zebrafish indicates specificity of arterial expression of each arterial enhancer. Stable transgenic zebrafish expressing the 15 strong arterial enhancer:GFP transgenes crossed with *tg(kdrl:HRAS-mCherry),* which expresses mCherry in all blood vascular ECs. Grey dashed box indicates region of zoom, da indicates dorsal aorta, cv indicates cardinal vein, dlav indicates dorsal longitudinal anastomotic vessel,

*Figure 3 continued on next page*

*Figure 3 continued*
pale pink filled arrows indicates intersegmental arteries, pale blue open arrows indicate intersegmental veins, * indicates expression in neural tube, and F1/2 indicates generation of embryo.

The online version of this article includes the following figure supplement(s) for figure 3:

**Figure supplement 1.** Analysis of enhancer activity in the 2–3 dpf ocular vasculature.

## In vivo enhancer activity does not always correlate with in silico predictions

We compared our results to a published genome-wide classification of EC regulatory elements, which used assessment of relative H3K27ac and acEP300 occupancy in freshly isolated HUVEC and HUAECs to classify enhancer regions as arterial-enriched, venous-enriched and common (arterial and venous enriched) regulatory elements (*Sissaoui et al., 2020*). Only 3/15 of our in vivo-validated arterial enhancers were classified as arterial enhancers using this assay, with 5/15 classified as common enhancers, 4/15 characterized as TSS, and 3/15 as uncalled (*Table 2*). A similar analysis of the eight previously identified arterial enhancers found none were classified as arterial enhancers (7/8 were marked as common enhancers and 1/8 was uncalled) (*Table 2*). Conversely, 10/19 of the regions that were inactive in our transgenic assays were classified as enhancers in the in vitro assay (8 as common enhancers, 2 as arterial enhancers) (*Table 2*). The low correlation between predicted activity and behaviour in transgenic assays suggests that in silico assessments using enhancer marks in cultured ECs alone may not strongly predict the ability of a putative enhancer region to independently direct gene expression nor the resultant specificity of this expression.

We also considered whether the use of vein-origin (HUVEC) and microvascular-origin (HMVEC-dBl-neo/ad) ECs in our analysis of human enhancer marks may have affected the accuracy of our putative enhancer selection by expanding our analysis to enhancer marks in arterial-origin ECs. However, analysis of enhancer marks in human aortic endothelial cells (HAEC and telo-HAECs) and human umbilical artery endothelial cells (HUAECs) showed a very similar pattern and identified the same set

| ENHANCER | 2dpf trunk vessels | | | | | 3dpf trunk vessels | | | | | | 3dpf ocular vessels | | | | | adult fin vessels | | |
|---|---|---|---|---|---|---|---|---|---|---|---|---|---|---|---|---|---|---|---|
| | DA | ISA | DLAV | *CV* | *NT* | DA | ISA | DLAV | *CV* | *ISV* | *NT* | NCA | NCA x | HA | *DCV* | *OV* | A | A sprout | *V* |
| *Cxcr4-194* | s | s | s | | w | s | s | s | *m\*\** | *m\*\** | m | s | s | s | *s* | *s* | s | s | *s\*\*\** |
| *Cxcr4+135* | | s | s | | | | s | s | | | | s | w | s | | | s | s | |
| *Cxcr4+151* | m | s | s | *w\** | w | w | s | s | | | | s | s | m | | | s | s | *s\*\*\** |
| *Efnb2-333* | s | s | m | | *s* | s | s | w | | | *s* | s | s | m | | | s | s | |
| *Efnb2-159* | | s | s | | | | s | s | | | | s | s | s | | | | | |
| *Efnb2-141* | s | | | | | s | | | | | | | | | | | s | | |
| *Efnb2-112* | m | m | m | *w* | | m | m | m | | | *w* | w | | | | | | | |
| *Gja4+50* | s | s | s | | | s | s | s | | | | s | | s | | | s | s | |
| *Unc5b+30* | s | m | m | *w\** | | s | w | | | | | s | | s | | | | | |
| *Unc5b+39* | | s | s | | *s* | | s | s | | *w* | *w* | s | | s | | | | s | |
| *Acvrl1+6* | s | w | | *w\** | *m* | s | | | *w\** | | *m* | m | m | s | *m* | *m* | | | |
| *Cxcl12+269* | | s | | | | | s | | | | *w* | w | | s | | | m | s | |
| *Gja5-78* | s | m | | | *w* | s | s | s | *w\** | *w* | | s | | s | | | s | w | |
| *Gja5-7* | s | w | | *w\** | | s | | | | | *w* | | | | | | m | w | |
| *Nrp1+78* | | s | s | *w\** | *w* | | s | s | | | *w* | | | | | | w | w | |

**Figure 4.** Summary of enhancer:GFP expression pattern in the vasculature of stable transgenic zebrafish lines. DA dorsal aorta, ISA intersegmental arteries, DLAV dorsal longitudinal anastomotic vessel, CV cardinal vein, ISV intersegmental veins, NT neural tube, NCA nasal ciliary artery, NCAx extends beyond NCA in direction of blood flow, HA hyaloid artery, DCV dorsal ciliary vein, OV optic vein. A fin artery, V fin vein. Letters "s" "m" and "w" equate to strong medium or weak relative expression, *restricted to distal regions, ** restricted to anterior regions, *** restricted to subset of ECs.

The online version of this article includes the following source data for figure 4:

**Source data 1.** Summary of enhancer:GFP expression pattern in the vasculature of stable transgenic zebrafish lines.

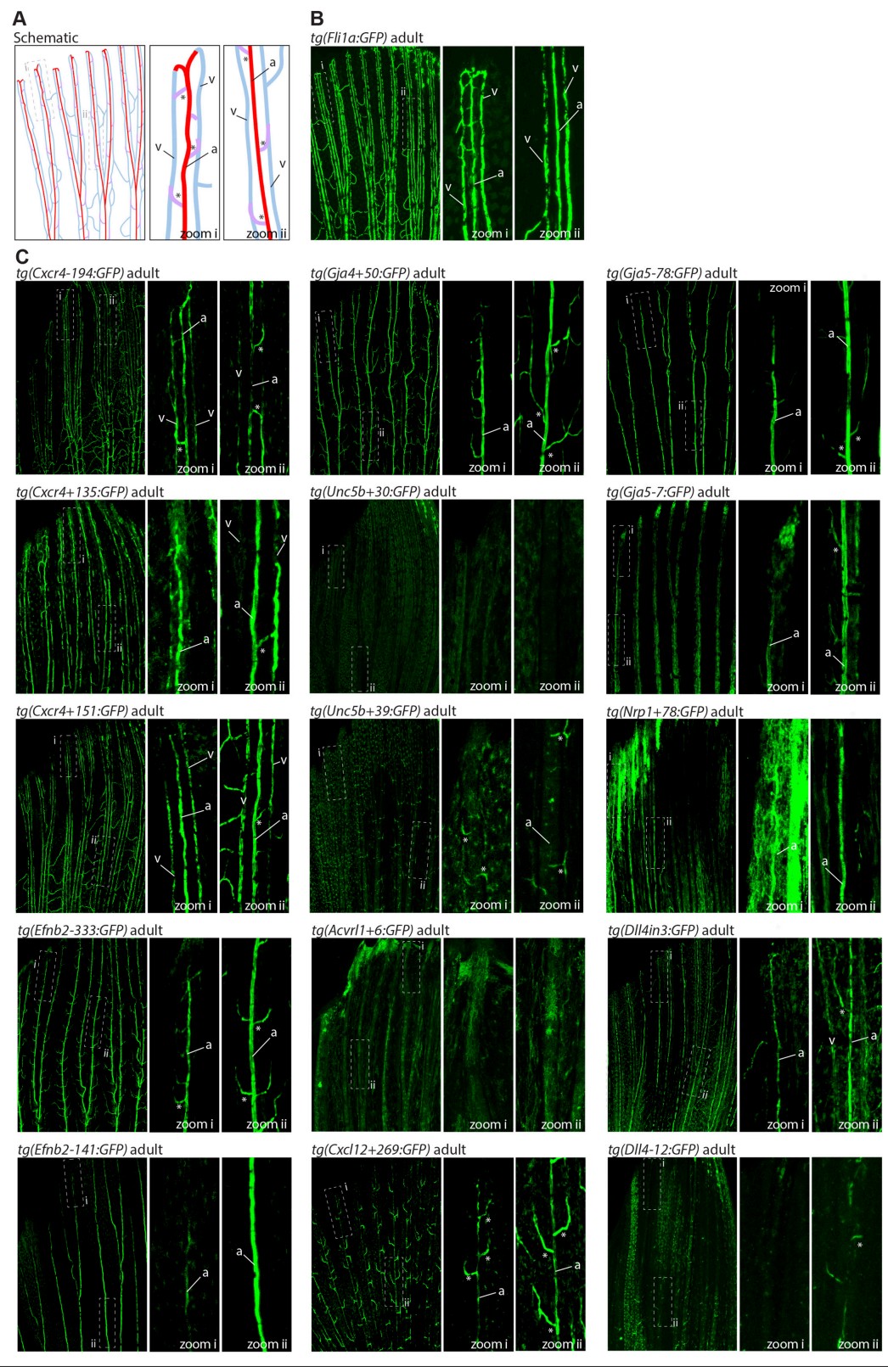

**Figure 5.** The majority of developmentally active arterial enhancers remain active in adult zebrafish fins. (**A**) Schematic drawing of the zebrafish fin vasculature adapted from *Xu et al., 2014*. (**B**) Expression pattern of the common EC marker line Fli1a:GFP in an adult fin. (**C**) Expression pattern of 13 novel arterial enhancer:GFP transgenes in adult fins, alongside previously identified arterial enhancers *Dll4in3* and *Dll4-12*. Grey dashed box

*Figure 5 continued on next page*

*Figure 5 continued*

indicates regions of zoom, a indicates fin artery, v indicates fin vein, * arterial sprout. See also *Figure 1—figure supplement 4*.

of putative enhancers as when HUVEC/HMVEC data was considered (*Figure 1—figure supplement 4*). This suggests that the arteriovenous original of cultured cells did not significantly influence putative enhancer marks, further emphasizing the challenges of using selective enhancer marks in such lines to predict expression patterns in vivo.

## Assessment of transcription factor motifs and binding patterns at arterial enhancers

This work so far has identified a cohort of enhancers able to drive strong gene expression selectively to arterial ECs. We next investigated whether this arterial expression pattern was associated with the binding of particular transcription factors. The ability of a transcription factor to bind an enhancer or promoter sequence is commonly established by identifying one or more binding motif(s) within the region of interest, and/or by observing direct binding to the region of interest by chromatin immunoprecipitation (ChIP) or similar methodologies. Here, we combined both approaches. First, we performed HOMER analysis to identify overrepresented motifs within the core regions of all 15 arterial enhancers identified here (≈250–400 bp, centred on enhancer marks and cross-species conservation) alongside all eight previously identified arterial enhancers (a total of 23). This HOMER motif analysis indicated the repeated presence of motifs for ETS (including EC-associated factors ETS1, ERG and ETV2), SOX (including EC-associated SOX17 and SOX7), FOX (including EC-associated FOXO1 and FOXO3), RBPJ (the transcriptional effector of NOTCH signalling), and MEF2 (including EC-associated MEF2A) (*Figure 7—figure supplement 1*). We next directly searched the sequences of all 23 core arterial enhancers to accurately determine the frequency of motifs for each of these transcription factors. In addition, we also looked for possible binding of other transcription factors previously implicated in arterial gene expression. The enhancer motif search used a combination of the JASPAR Transcription Factors Track Settings (TFTS) on the UCSC Genome Browser and hand annotation using previously defined consensus motifs (*Figure 7—figure supplement 1*). Because the level of conservation of motifs can often be an indication of their importance to enhancer activity, we classified each motif into three categories: strongly conserved (motif conserved to the same depth of the surrounding sequence), weakly conserved (motif conserved in orthologous human enhancer but not to the same depth as the surrounding sequence), and not conserved (motif is not conserved within the orthologous human sequence). In parallel, we compared this motif analysis with a variety of published endothelial ChIP-seq and CUT&RUN datasets to determine where there was evidence of direct binding for each of these transcription factors at each enhancer (see *Figure 7* and *Figure 8*, *Figure 7—figure supplements 2–6*, *Figure 8—figure supplements 1 and 2*, *Figure 9*).

## ETS, SOXF, and FOX binding is a common but not unique occurrence at arterial enhancers

Our motif analysis revealed near-ubiquitous binding sites for ETS, SOXF and FOX transcription factors at all arterial enhancers (*Figure 7*, *Figure 7—figure supplements 2–6*, *Figure 9*). For ETS, 23/23 arterial enhancers contained at least one strongly conserved motif. Confirming this motif identification, 18/23 of these enhancers also directly bound ERG and ETS1, two very common EC-associated ETS factors (*Figure 8*, *Figure 8—figure supplement 1*). For FOX, 22/23 arterial enhancers contained conserved FOX motifs (22 strongly conserved) with 13/23 directly binding FOXO1 (*Figure 8*, *Figure 8—figure supplement 1*). For SOXF, 22/23 arterial enhancers contained conserved motifs (21 strongly conserved). Initial comparison of our enhancer cohort with publicly available information on SOX7 binding (from *Overman et al., 2017*, which used ChIP-seq in HUVECs over-expressing SOX7-mCherry) showed no overlap. However, only 6% of SOX7-mCherry peaks overlapped with EC-associated enhancers or TSS marks and only 4% correlated with ERG binding, an ETS family transcription factor strongly associated with EC gene transcription (*Figure 8—figure supplement 3A*, ERG and enhancer mark data from *Sissaoui et al., 2020*). Further, no SOX17 ChIP-seq in ECs has been published, despite SOX17 being most closely associated with arterial identity. We therefore

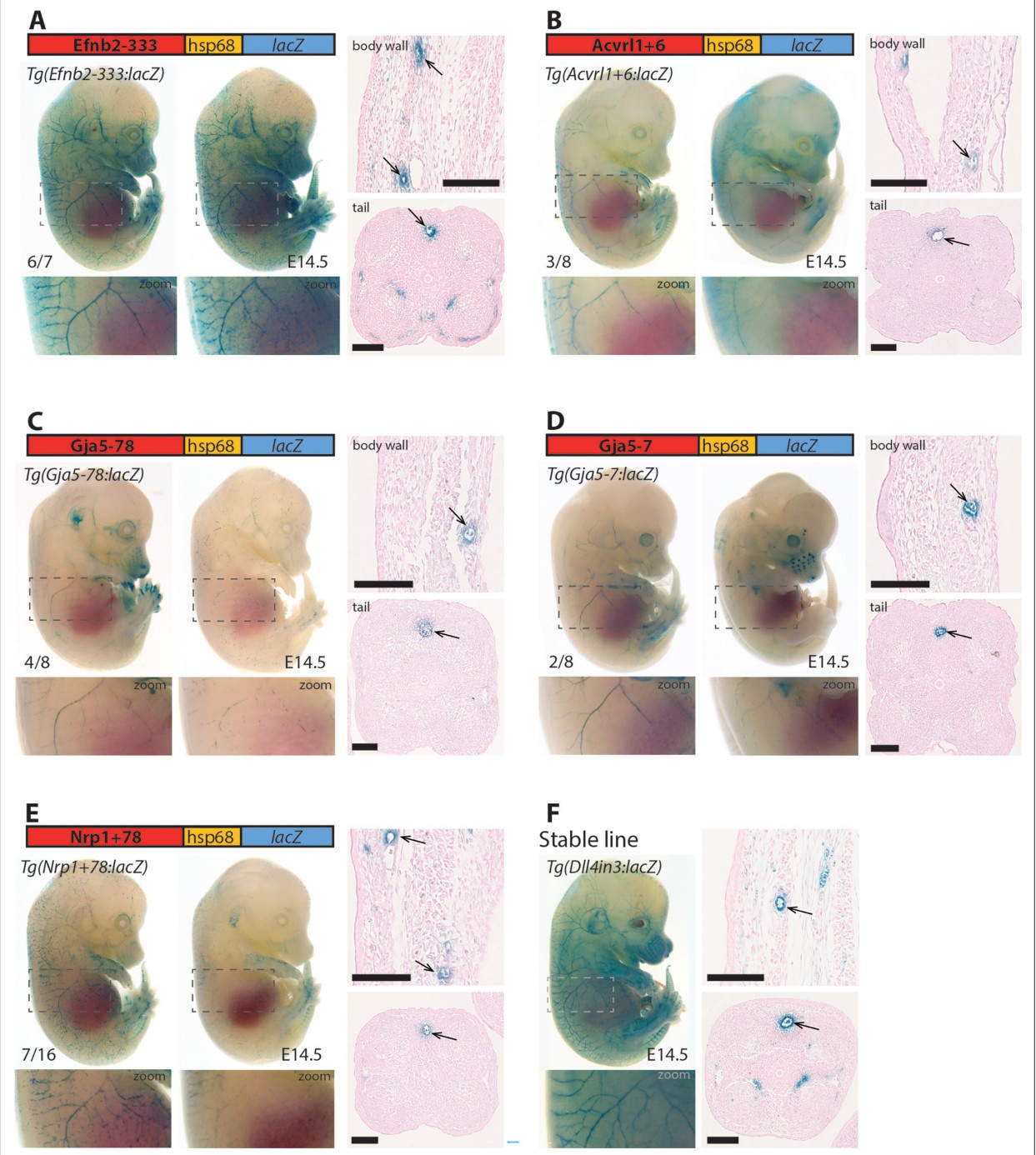

**Figure 6.** Five putative enhancers direct arterial expression in the vasculature of E14.5 transgenic mouse embryos. (**A–E**) Two representative F0 embryos expressing each tested putative enhancer alongside a schematic of the transgene and two transverse sections through the embryo body wall and tail. Numbers in bottom left indicate embryos with arterial lacZ/total transgenic embryos. Grey dashed boxes indicate region in zoom, arrow indicates artery. (**F**) shows a representative E14.5 embryo from a stable transgenic line expressing the arterial *Dll4in3:lacZ* transgene alongside similar transverse sections through the embryo body wall and tail. Black line = 100 um.

The online version of this article includes the following figure supplement(s) for figure 6:

**Figure supplement 1.** All transgenic embryos expressing the *Efnb2-37:lacZ* transgene at E14.5 Grey dashed boxes indicate region in zoom.

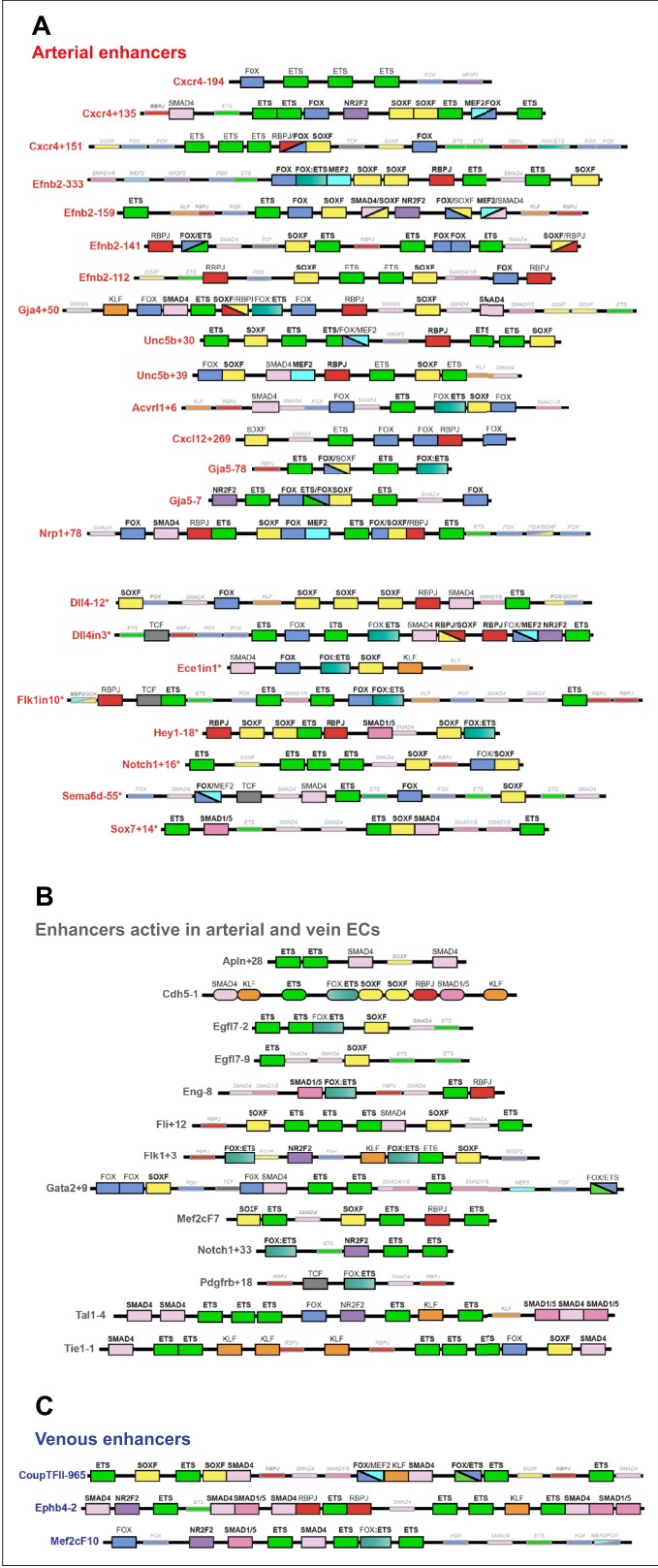

**Figure 7.** Schematics summarizing the transcription factor motifs found within each arterial (**A**), pan-EC (**B**), and venous (**C**) enhancer. All enhancers shown in 3′–5′ orientation relative to the arterial gene TSS. Deep black-lined rectangle boxes indicate strongly conserved motifs for transcription factors (conserved at the same depth as the surrounding enhancer sequence), shallow grey-lined boxes/text indicate weakly conserved motifs (conserved

*Figure 7 continued on next page*

*Figure 7 continued*

between mouse and human enhancer sequence but not at the same depth as the surrounding sequence), and rounded boxes mark motifs in enhancers conserved only human-mouse. Bold transcription factor names indicate places where ChIP-seq (or similar analysis) confirms binding. See *Figure 7—figure supplements 2–6* for annotated sequences. Arterial enhancers listed with * are previously published (as detailed in *Payne et al., 2024*), genome locations for each enhancer are provided in *Table 2—source data 2*. Distances between motifs are representative but not scaled.

The online version of this article includes the following figure supplement(s) for figure 7:

**Figure supplement 1.** Transcription factor motifs associated with arterial enhancers.

**Figure supplement 2.** Sequences of core enhancers (3' to 5' orientation) listed in *Figure 7* alongside annotated transcription factor binding motifs.

**Figure supplement 3.** Sequences of core enhancers (3' to 5' orientation) listed in *Figure 7* alongside annotated transcription factor binding motifs.

**Figure supplement 4.** Sequences of core enhancers (3' to 5' orientation) listed in *Figure 7* alongside annotated transcription factor binding motifs.

**Figure supplement 5.** Sequences of core enhancers (3' to 5' orientation) listed in *Figure 7* alongside annotated transcription factor binding motifs.

**Figure supplement 6.** Sequences of core enhancers (3' to 5' orientation) listed in *Figure 7* alongside annotated transcription factor binding motifs.

---

performed an assessment of SOX17 binding with antibodies against the endogenous protein using CUT&RUN in HUVECs. In this analysis, 75% of SOX17 binding peaks overlapped with EC enhancer/promoter marks (*Sissaoui et al., 2020*), 73% overlapped with ERG binding (*Sissaoui et al., 2020*), and HOMER analysis identified the SOX17 consensus motif as the most significantly enriched motif (*Figure 7—figure supplement 3B and C*). Assessment of our 23 arterial enhancer cohort found called SOX17 peaks at 20/23 arterial enhancers, in every case correlating with the presence of strongly conserved SOXF motifs (*Figure 8*, *Figure 8—figure supplement 1*, *Figure 9*). We also considered whether the pattern of SOX17 binding was different to SOX7 and SOX18, proteins with similar binding motifs but different expression profiles within the vasculature (*Zhou et al., 2015*). For this, we again used CUT&RUN in HUVECs with antibodies against the endogenous SOX7 and SOX18 proteins. In these assays, 91% of SOX17 peaks were also bound by SOX7 and/or SOX18 (86% shared with SOX7, 71% with SOX18, 66% with both) (*Figure 8—figure supplement 3*). For our arterial enhancers, all 20 SOX17-bound enhancers were also bound by either SOX7 (9/23), SOX18 (2/23) or both SOX7 and SOX18 (9/23) (*Figure 4*, *Figure 8*, *Figure 8—figure supplement 1*). These data therefore suggest that binding of the arterial-enriched SOX17 was not a specific event at arterial-selective enhancers and that other SOXF proteins can also recognize and bind the SOX motifs within these enhancers.

Looked at in isolation, this analysis would strongly suggest that ETS, FOX, and SOXF transcription factors work together to direct arterial-specific gene expression. However, although our enhancers are all specific or highly enriched in arterial ECs, this analysis cannot by itself distinguish between factors that specifically direct arterial expression, and those required for endothelial gene expression more generally. Consequently, to determine if these common ETS, SOXF and FOX binding patterns were unique to arterial enhancers, we expanded our analysis to 16 in vivo-validated endothelial enhancers that were not selectively active in arterial ECs (*Figure 7B and C*, *Figure 7—figure supplements 5 and 6* and *Figure 8—figure supplement 2*, *Figure 9*). As assessed in transgenic mouse embryos at mid-late gestation, 13 of these enhancers drove relatively equal expression in arterial and venous ECs (pan-EC enhancers), while the activity of the other three was vein-enriched and artery-excluded (vein enhancers) (*Payne et al., 2024*). Unsurprisingly, given their known role in general endothelial gene expression and identity, binding of ETS factors was ubiquitous at pan-EC enhancers and venous enhancers in addition to arterial enhancers. However, despite the association of SOXF factors with arterial identity, 12/13 pan-EC enhancers also directly bound SOXF factors (11/13 bound by SOX17, 11/13 by SOX7, and 9/13 by SOX18), with 9/13 containing strongly conserved SOXF motifs (*Figure 8—figure supplement 2*, *Figure 9*). The binding patterns at venous enhancers were harder to interpret due to a limited number of well-validated enhancers in this category and the activity of the *CoupTFII-965* enhancer in lymphatic ECs in addition to veins. While the *Ephb4-2* or *Mef2cF10*

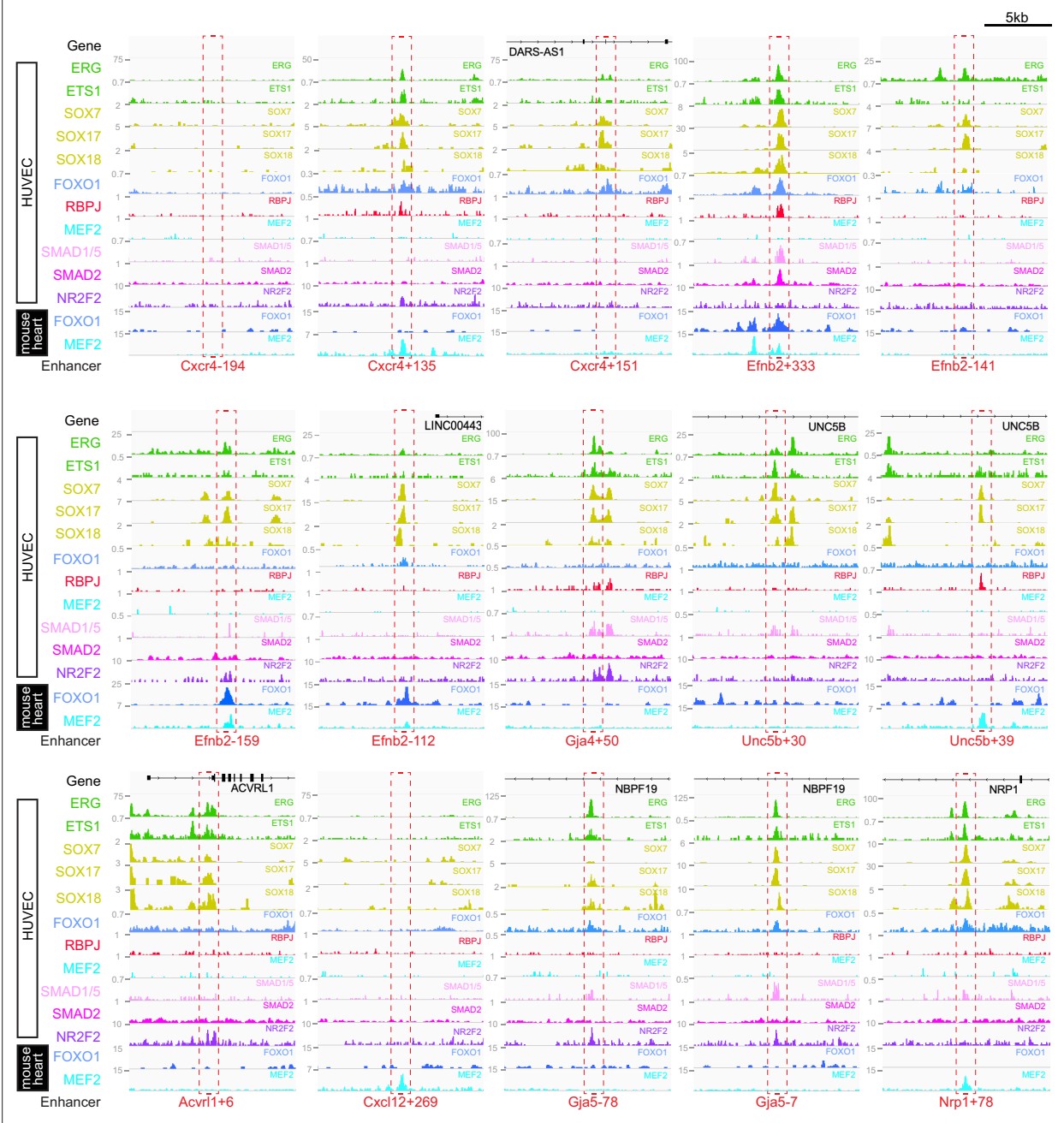

**Figure 8.** Binding patterns of 11 vascular-associated transcription factors around each arterial enhancer. Red dashed box indicates arterial enhancer region. Tracks show ChIP-seq/CUT&RUN signal for ERG (*Sissaoui et al., 2020*), ETS1 (*Chen et al., 2017*), SOX7, SOX17 and SOX18 (this paper), FOXO1 (*Andrade et al., 2021*), RBPJ (*Wang et al., 2019*), MEF2C (*Maejima et al., 2014*), SMAD1/5 (*Morikawa et al., 2011*), SMAD2 (*Chen et al., 2019*), and NR2F2 (*Sissaoui et al., 2020*) in HUVECs, alongside FOXO1 (*Sissaoui et al., 2020*), and MEF2A (*Akerberg et al., 2019*) in adult mouse hearts.

The online version of this article includes the following figure supplement(s) for figure 8:

**Figure supplement 1.** Genomic regions around eight previously described arterial enhancers (red dashed box) alongside tracks showing ChIP-seq/CUT&RUN signal for ERG (*Sissaoui et al., 2020*), ETS1 (*Chen et al., 2017*), SOX7, SOX17, and SOX18 (this paper), FOXO1 (*Andrade et al., 2021*), RBPJ (*Wang et al., 2019*), MEF2C (*Maejima et al., 2014*), SMAD1/5 (*Morikawa et al., 2011*), SMAD2 (*Chen et al., 2019*), and NR2F2 (*Sissaoui et al., 2020*) in HUVECs, alongside FOXO1 (*Sissaoui et al., 2020*) and MEF2A (*Akerberg et al., 2019*) in adult mouse hearts.

**Figure supplement 2.** Genomic regions around 13 previously described pan-EC enhancers (grey dashed box) and 3 previously described vein enhancers (dark blue dashed box) alongside tracks showing ChIP-seq/CUT&RUN signal for ERG (*Sissaoui et al., 2020*), ETS1 (*Chen et al., 2017*), SOX7, SOX17, and SOX18 (this paper), FOXO1 (*Andrade et al., 2021*), RBPJ (*Wang et al., 2019*), MEF2C (*Maejima et al., 2014*), SMAD1/5 (*Morikawa et al.,*

*Figure 8 continued on next page*

*Figure 8 continued*

*2011*), SMAD2 (*Chen et al., 2019*), and NR2F2 (*Sissaoui et al., 2020*) in HUVECs, alongside FOXO1 (*Sissaoui et al., 2020*) and MEF2A (*Akerberg et al., 2019*) in adult mouse hearts.

**Figure supplement 3.** Assessment of SOX7, SOX17 and SOX18 CUT&RUN.

**Figure supplement 4.** Gene expression patterns of SOX and FOX transcription factors in ECs from different scRNA-seq datasets.

vein enhancers contained no SOXF motifs, the *CoupTFII-965* enhancer contained strongly conserved SOXF motifs and bound all SOXF proteins (*Figure 8—figure supplement 2*, *Figure 9*). Consequently, while the enhancers of some venous genes may lack sensitivity to SOXF factors, our data suggests that SOXF binding is not a unique feature of arterial-restricted gene expression. These observations align with previous studies showing roles for SOXF factors in vasculogenesis and angiogenesis (*Lilly et al., 2017*; *Kim et al., 2016*; *Lee et al., 2014*), and with the expression of SOXF factors throughout the vascular plexus during arterial-venous differentiation in the embryonic heart (*Figure 8*, *Figure 8—figure supplement 4* and *Chiang et al., 2017*; *Sharma et al., 2017*) and postnatal retina (*Zhou et al., 2015*). This also agrees with the severe vascular phenotypes seen after compound loss of SOXF factors, which include EC hyperplasia, loss of angiogenic markers, and inhibited arteriovenous differentiation (*Kim et al., 2016*; *Zhou et al., 2015*).

Our findings were somewhat less clear for FOX transcription factors. 9/13 pan-EC enhancers contained some kind of FOX motif (compared to 22/23 arterial enhancers). However, the only FOX motif within 6 of these pan-EC enhancers was a composite part of a FOX:ETS motif (a vasculogenic-associated element whose FOX component is often fairly degenerate) *Figures 7 and 8*, *Figure 7—figure supplements 1–6*; *Figure 7—figure supplements 2–6*; *Figure 7—figure supplements 4–6*; *Figure 7—figure supplements 3–6*; *Figure 7—figure supplements 3–5*; *Figure 9* This leaves only 3/13 pan-EC enhancers containing independent FOX motifs, compared to 20/23 arterial enhancers, suggesting that FOX binding may be enriched among our arterial enhancer cohort. While this aligns with previous observations linking FOXC1 and FOXC2 with arterial differentiation expression (*Seo et al., 2006*), neither are highly expressed in developing coronary arteries nor commonly identified as arterial-enriched in single cell transcriptomics in developing mouse embryos (*Figure 8*, *Figure 8—figure supplement 4* and *Hou et al., 2022*; *Chen et al., 2024*). In addition to FOXC1/2, FOXO1 is also expressed widely throughout the endothelium and directly bound many of our arterial enhancers, but direct links between deletion or constitutive activation of FOXO1 and changes in arterial differentiation have not yet been reported (e.g.*Wilhelm et al., 2016*).

## MEF2 binding occurs at a subset of arterial enhancers and may be associated with sprouting

MEF2 factor binding was overrepresented in arterial enhancers compared to pan-EC enhancers, although it was seen at only a minority of enhancers. In total, 9/23 arterial enhancers contained conserved MEF2 motifs (8 strongly conserved) compared to only 1/13 pan-EC enhancers. This arterial-skewed pattern was repeated in assays of MEF2 factor binding, with direct MEF2 binding found at 8/23 arterial enhancers and only 2/13 pan-EC enhancers (*Figure 8*, *Figure 8—figure supplements 1 and 2*, *Figure 9*). Given the known role of MEF2 factors in angiogenic sprouting and the close link between angiogenesis and arterial differentiation (*Sacilotto et al., 2016*; *Pitulescu et al., 2017*), it is possible that MEF2 factors are regulating gene expression in response to angiogenic rather than arterial cues at these enhancers. This has been already shown for the *Dll4in3* enhancer, where loss of MEF2 binding ablated enhancer activity in angiogenic ECs but not in mature arterial ECs (*Sacilotto et al., 2016*). Supporting this hypothesis, MEF2-bound enhancers were associated with genes expressed in early 'pre-arterial' EC and/or involved in angiogenesis (*Cxcr4*, *Efnb2*, *Nrp1*, *Unc5b*, *Flk1*, and *Dll4*) (*Hou et al., 2022*; *Raftrey et al., 2021*; *Figure 9*). To determine if there was a relationship between MEF2 motif/binding and expression pattern, we grouped the enhancers according to zebrafish expression profiles in the trunk and looked at the resultant clusters of TFs (*Figure 9*). Strikingly, all five arterial enhancers restricted to the intersegmental vessels were MEF2-bound, indicating a possible angiogenic role as these vessels form by angiogenic sprouting. In addition, the MEF2-bound *Efnb2-333* is active in both compartments in a similar pattern to the Dll4in3. However, MEF2 factors are also associated with transcriptional activation downstream of shear stress, which is a known effector of

**A**

| ENHANCER | In vivo class | ETS | SOXF | FOXO | FOX:ETS | RBPJ | MEF2 | SMAD4 M | SMAD1/5 M | ChIP-seq | TCF/LEF | NR2F2 | KLF4 |
|---|---|---|---|---|---|---|---|---|---|---|---|---|---|
| Cxcr4-194 | Arterial* | M | | M | | | | | | | | m | |
| Cxcr4+135 | Arterial | M+C | M+C 7,17 | M+C | ADJACENT | m+C | M+C | M | | | | M+C | c |
| Cxcr4+151 | Arterial | M | M+C 7,17 | M+C | m | M | | | | | m | | |
| Efnb2-333 | Arterial | M+C | M+C 7,17,18 | M+C | M | M+C | M+C | m | m | C 1/5,2 | | m | |
| Efnb2-159 | Arterial | M+C | M+C 7,17 | M+C | | | M | M | | C 1/5 | | M+C | m |
| Efnb2-141 | Arterial | M+C | M+C 7,17 | M+C | OVERLAP | | M | m | | | | | |
| Efnb2-112 | Arterial | M | M+C 7,17,18 | M+C | | | M | m | m | | | | |
| Gja4+50 | Arterial | M+C | M+C 7,17 | M | M | M+C | | M | m | C 1/5 | | c | M |
| Unc5b+30 | Arterial | M+C | M+C 7,17 | M | OVERLAP | M | M | | | | | m | |
| Unc5b+39 | Arterial | M | M+C 7,17 | M | | M+C | M+C | M | | | | | m |
| Acvrl1+6 | Arterial | M+C | M+C 7,17,18 | M | M | m | | M | m | | | c | m |
| Cxcl12+269 | Arterial | M | M | M | | M | c | m | | | | | |
| Gja5-78 | Arterial | M+C | M+C 17,18 | M+C | M | m | | | | | | c | |
| Gja5-7 | Arterial | M+C | M+C 7,17,18 | M+C | OVERLAP | | | m | | C 1/5 | | M+C | |
| Nrp1+78 | Arterial | M+C | M+C 7,17,18 | M+C | ADJACENT | M | M+C | M | | C 1/5 | | | |
| DLL4-12* | Arterial | M+C | M+C 7,17 | M+C | | | M | M | m | | | | m |
| DLL4in3* | Arterial | M+C | M+C 17,18 | M | M | M+C | M+C | M | | | M | M+C | |
| ECE1in1* | Arterial | M+C | M+C 7,17,18 | F:E ONLY | M | | | M | | C 1/5 | | c | M |
| Flk1in10* | Arterial | M+C | m | M+C | M | M | m+C | m | m | | M | | m |
| Hey1-18* | Arterial | M+C | M+C 7,17,18 | F:E ONLY | M | M+C | | m | M | C 1/5,2 | | c | m |
| NOTCH1+16* | Arterial | M+C | M+C 7,17 | M | | m | | m | m | | | c | m |
| Sema6d-55* | Arterial | M+C | M+C 7,17,18 | M+C | | | M | M | | | M | | |
| SOX7+14* | Arterial | M+C | M+C 7,17,18 | c | | | | M | M | C 1/5,2 | | c | |
| Apln+28* | Pan-EC | M+C | m+C 7,17,18 | | | | | M | | | | c | m |
| Cdh5-1* | Pan-EC | M+C | M+C 7,17 | F:E ONLY | M | M | | M | M | | | | M |
| Egfl7-2* | Pan-EC | M+C | M+C 7,17,18 | F:E ONLY | M | | | m | m | C 1/5,2 | | c | m |
| Egfl7-9* | Pan-EC | M+C | M+C 7,17,18 | c | | | | m | | C 1/5,2 | | | |
| Eng-8* | Pan-EC | M+C | c 7 | F:E ONLY | M | M | | m | M | | | c | m |
| Fli1+12* | Pan-EC | M+C | M+C 7,17,18 | m | | m | | M | | | | | |
| Flk1+3* | Pan-EC | M+C | M+C 7,17,18 | F:E ONLY | M | m | | | | | | M+C | M |
| Gata2+9* | Pan-EC | M+C | M+C 7,17,18 | M | OVERLAP | | m+C | M | m | | M | c | |
| Mef2cF7* | Pan-EC | M+C | M+C 7,17,18 | c | | M | | m | | | | c | m |
| Notch1+33* | Pan-EC | M+C | c 7, 17, 18 | F:E ONLY | M | | | | | c | | M+C | |
| Pdgfrb+18* | Pan-EC | F:E ONLY | | F:E ONLY | M | m | | m | | | M | | |
| Tal1-4* | Pan-EC | M+C | M+C17 | M | | | c | M | M | C 1/5,2 | | M | M |
| Tie1-1* | Pan-EC | M+C | M+C 7,17,18 | M | ADJACENT | M | | M | | C 2 | | c | M |
| CoupTFII-965* | Venous | M+C | M+C 7,17,18 | M+C | OVERLAP | m+C | M | M | m | C 1/5,2 | | | M |
| Ephb4-2* | Venous | M+C | c 7, 17, 18 | | | | M | M | M | C 1/5,2 | | M+C | M |
| Mef2C F10* | Venous | M+C | | M | M | | m | M | M | | | M+C | |

**B**

| | | | TF motif patterns | | | | | | | | | | 3 dpf trunk vessels | | | | | | |
|---|---|---|---|---|---|---|---|---|---|---|---|---|---|---|---|---|---|---|---|
| ETS | SOXF | FOXO | FOX:ETS | RBPJ | MEF2 | SMAD 4 | 15 | ChIP | TCF | NR2F2 | KLF4 | DA | ISA | DLAV | CV | ISV | NT | |
| M | | M | | | | | | | | m | | s | s | s | m** | m** | m | Cxcr4-194 |
| M+C | M+C 17,18 | M | M | M+C | M+C | M | | | M | M+C | | s | s | s | | w | | DLL4in3* |
| M+C | M+C 7,17 | M | M | M+C | | M | m | 15 | | c | M | s | s | s | | | | Gja4+50 |
| M+C | M+C 17,18 | M+C | M | | | m | | | | c | | s | s | s | w* | w | | Gja5-78 |
| M | M+C 7,17,18 | M+C | | | M | | m | m | | | | m | m | m | | w | w | Efnb2-112 |
| M+C | M+C 7,17,18 | M+C | M | M+C | M+C | m | m | 15 2 | | m | | s | s | w | | | s | Efnb2-333 |
| M+C | M+C 7,17 | M | OLAP | M | M | | | | | m | | s | w | | | | | Unc5b+30 |
| M+C | M+C 7,17 | M+C | | M | | M | m | | | | m | s | w | | w* | | s | DLL4-12* |
| M+C | M+C 7,17,18 | M+C | OLAP | | | m | | 15 2 | | M+C | | s | | | | w | | Gja5-7 |
| M+C | M+C 7,17 | M+C | OLAP | M | | m | | | | | | s | | | | | | Efnb2-141 |
| M+C | M+C 7,17,18 | M | M | m | | M | m | | | c | m | s | | | w* | | m | Acvrl1+6 |
| M | M+C 7,17 | M+C | | M | | M | m | | m | | | w | s | s | | | | Cxcr4+151 |
| M | M | M | | M | c | m | | | | | | s | | | | | w | Cxcl12+269 |
| M+C | M+C 7,17 | M+C | ADJ | m+C | M+C | M | | | | M+C | c | s | s | | | | | Cxcr4+135 |
| M+C | M+C 7,17 | M+C | | M | M+C | M | | 15 2 | | M+C | m | s | s | | | | | Efnb2-159 |
| M | M+C 7,17 | M | | M+C | M+C | M | | | | | m | s | s | | w | w | | Unc5b+39 |
| M+C | M+C 7,17,18 | M+C | ADJ | M | M+C | M | | 15 2 | | | | s | s | | | w | | Nrp1+78 |

**Figure 9.** Summary of transcription factor motif and binding patterns at arterial, pan-EC and venous enhancers (**A**), and relative to different expression patterns within the arterial vasculature (**B**). (**A**) All known (e.g. published) endothelial enhancers with adequately described expression patterns in transgenic mouse embryos were analysed to determine occurrence of selected TF motifs and direct binding. See *Figure 7—figure supplement 1* for TF motif information, *Figure 7—figure supplements 2–6* for annotated enhancer sequences and *Figure 8* and *Figure 8—figure supplements 1 and*

*Figure 9 continued on next page*

*Figure 9 continued*

*2* for TF binding peaks. Enhancers in bold were identified in this paper, those with * are previously published (as detailed in [1]), genome locations for each enhancer is provided in *Table 2—source data 2*. (**B**) TF binding patterns for each arterial enhancer grouped by expression patterns in the 3 dpf zebrafish trunk. DA dorsal aorta, ISA intersegmental arteries, DLAV dorsal longitudinal anastomotic vessel, CV cardinal vein, ISV intersegmental veins, NT neural tube, NCA nasal ciliary artery, NCAx extends beyond NCA in direction of blood flow, HA hyaloid artery, DCV dorsal ciliary vein, OV optic vein. A fin artery, V fin vein. Letters s m w equate to strong medium or weak relative expression, * restricted to distal regions, ** restricted to anterior regions, *** restricted to subset of ECs.

The online version of this article includes the following source data for figure 9:

**Source data 1.** Summary of transcription factor motif and binding patterns atarterial, pan-EC and venous enhancers, and relative to different expressionpatterns within the arterial vasculature.

**Source data 2.** TF binding patterns for each arterial enhancer grouped by expression patterns in the 3 dpf zebrafish trunk.

*Efnb2* and other arterial gene expression (*Hwa et al., 2017*; *Lu et al., 2021*), and the MEF2-bound *Unc5b+30* enhancer was preferentially active in the dorsal aorta only (*Figures 2–4*). Further studies would therefore be required to definitely link MEF2 binding at these enhancers to either an angiogenic or sheer stress response.

## RBPJ binding indicates a role for Notch in transcription of arterial genes

RBPJ is the transcriptional effector of the Notch pathway, complexing with the Notch intracellular domain (NICD) and the co-activator MAML in order to directly bind DNA and activate transcription (*Bray, 2006*). Strongly conserved RBPJ motifs were found in 14/23 arterial enhancers and 4/13 pan-EC enhancers, while direct RBPJ binding was confirmed at 6/23 arterial enhancers only (*Figures 7 and 8*, *Figure 8—figure supplements 1–3*, *Figure 9*). RBPJ motifs are relatively short and share close similarity to the ETS motifs (consensus TGGGAA vs. HGGAAR), potentially explaining the discrepancy between motif and direct binding. Direct RBPJ binding to arterial enhancers has previously only been reported for genes in the Notch pathway, a fact that supported the hypothesis that Notch does not directly induce arterial differentiation through gene activation but instead by reducing their MYC-dependent metabolic and cell-cycle activities (*Luo et al., 2021*). However, here we found good evidence for RBPJ binding at enhancers for *Cxcr4*, *Efnb2*, *Gja4*, and *Unc5b*, suggesting that Notch may directly influence at least some arterial identity genes. In most cases, these genes also contained additional enhancers not directly bound by RBPJ, potentially providing an explanation for the maintenance of some arterial gene expression in the absence of RBPJ/Notch (*Luo et al., 2021*). Additionally, previous studies on the *Dll4in3* enhancer found that loss of RBPJ (and Notch signalling) did not affect enhancer or gene expression unless SOXF factors were also perturbed (*Sacilotto et al., 2013*). Cooperation between SOXF and Notch may also partly explain how the widely expressed SOXF factors enact arterial-specific gene activation, although SOX factors bound arterial enhancers more commonly than RBPJ. An alternative explanation may be that these results instead reflect the known involvement of RBPJ/Notch in angiogenesis. However, whilst all RBPJ-bound arterial enhancers were active in the sprout-formed intersegmental vessels in zebrafish (*Figure 9*), neither RBPJ binding or RBPJ motifs were ubiquitous in intersegmental-expressed enhancers.

## No other arterial or venous-related transcription factors are commonly present or excluded at our arterial enhancers

Lastly, we investigated potential roles for SMADs (transcription factors downstream of TGFβ/BMP signalling), TCF7/TCF7L1/TCFL2/LEF1 (transcription factors downstream of canonical WNT signalling), and KLF4 (downstream of laminar shear stress). Although the binding motifs for these factors were not overrepresented in our arterial cohort as assessed by HOMER analysis, these pathways have all previously been implicated in arterial gene expression. We also looked for evidence of NR2F2/COUP-TFII binding, a vein and lymphatic-specific transcription factor previously implicated in both activation of venous genes and repression of arterial/Notch genes. This analysis found little evidence supporting a broad role for any of these pathways in arterial gene expression nor a link to any particular expression type within the arterial tree (*Figures 7 and 8*, *Figure 7—figure supplements 2–6*, *Figure 8—figure supplements 1 and 2*, *Figure 9*). NR2F2 motifs were seen in 7/23 arterial enhancers (but only strongly

conserved in 4/23) and 3/13 pan-EC enhancers. Largely uncorrelated ChIP-seq peaks were seen at 11/23 arterial enhancers, and at 8/13 pan-EC enhancers. Strongly conserved KLF4 motifs were only seen in 2/23 arterial and 4/13 pan-EC enhancers but none correlated with KLF4 binding. Strongly conserved TCF/LEF motifs were only found in 3/23 arterial enhancers and 2/13 pan-EC enhancers. SMAD1/5-SMAD4 factors downstream of BMP signalling have been previously associated with the expression of venous genes including *Nr2f2* and *Ephb4* (**Neal et al., 2019**; **Stewen et al., 2024**), and all three vein enhancers contained multiple motifs for SMAD factor binding (with 2/3 also directly binding SMAD1/5 and SMAD2 in HUVECS after BMP9/TGFβ stimulation) (**Figures 7 and 8**, **Figure 8—figure supplement 2**, **Figure 9**). However, here we found that SMAD binding also occurred at arterial enhancers, with 12/23 containing strongly conserved motifs, 8/23 directly binding SMAD1/5 of which three also bound SMAD2. This agrees with the lack of venous-specificity reported for phosphorylated SMAD1/5, which led to the supposition that addition factors work alongside SMAD1/5 to regulate vein specification (**Neal et al., 2019**). An arterial role for SMADs is not without precedent, particularly downstream of TGFβ (e.g.**Ola et al., 2018**; **Chavkin et al., 2022**; **Roman and Hinck, 2017Roman and Hinck, 2017**; **Daems et al., 2024**). While Tie2:Cre-mediated excision of SMAD4 (effectively knocking out all canonical BMP/TGFβ signalling) did not obviously affect arterial differentiation at E9.5, an earlier or later role cannot be dismissed, as Tie2:Cre becomes active after vasculogenic-driven arterial differentiation occurs, and vein-related lethality occurs in these embryos by E10.5 (**Neal et al., 2019**), prior to most vein/capillary-to-arterial EC differentiation.

## Discussion

Recent years have brought a new appreciation of the vein/capillary origin of most arterial ECs, and an increasing interest in arterialization as a therapeutic aim of regenerative medicine. However, the transcriptional pathways driving arterial differentiation are still incompletely understood. Many factors have contributed to this, including a focus on the Notch signalling pathway and a lack of characterized arterial enhancers for most key arterial genes. The latter has resulted in regulatory pathways being linked to arterial gene expression through proposed binding at promoter regions, although these elements are often poorly characterized or unsupported by functional data (e.g. binding motifs located kilobases away from TSS at regions without promoter or enhancer marks, transcription factor binding not verified by available ChIP-seq datasets). As well as the potential for incorrect assumptions, a reliance on poorly defined enhancer/promoter regions prevents further research building on these initial observations, for example, by looking for associated motifs to identify combinatorial, synergistic, and antagonistic factors or to link with newly discovered pathways or transcription factors. In this paper, we sought to generate a useful and accessible cohort of arterial enhancers with which to study arterial transcriptional regulation more effectively. Alongside *Dll4*, *Notch1,* and *Hey1* (all genes with previously described enhancers included in our analysis), the eight arterial genes focused on here represent the majority of genes used to define arterial identity in single cell transcriptomics in mice and humans (e.g. **Hou et al., 2022**; **McCracken et al., 2022**; **Raftrey et al., 2021**; **Chen et al., 2024**). Further, our choice of targets included genes with essential and well-studied roles in arterial differentiation (e.g. *Efnb2*), implicated in arteriovenous malformations in humans (e.g. *Acvrl1*), associated with processes important for regeneration (*Cxcr4* and *Cxcl12*; **Das et al., 2019**), or commonly used as arterial markers in animal models (e.g. *Gja5*). Thus, our hope was to identify a cohort of arterial enhancers likely to be directly targeted by arterial lineage specification and differentiation factors that represent key stages of arterial development of interest to a wide range of researchers and that we can easily link to previous observations on arterial development in animal models of gene depletion.

Analysis of single-cell transcriptomic data has indicated that arterial ECs can be further subdivided into two groups, reflecting maturity but also potentially slightly different developmental trajectories (**Hou et al., 2022**; **Raftrey et al., 2021**). The genes studied here cover both subgroups, with *Acvrl1*, *Cxcl12*, *Gja5,* and *Nrp1* primarily restricted to the mature arterial EC subgroup, while *Cxcr4*, *Efnb2*, *Gja4,* and *Unc5b* were also expressed in the less mature/arterial plexus/pre-arterial EC subgroup (**Hou et al., 2022**; **Raftrey et al., 2021**). Although we saw no obvious differences in transcription factor motif and binding between the two sets overall, the genes expressed in both immature and mature subgroups tended to have multiple enhancers with differential expression patterns: there are four *Efnb2* enhancers, of which *Efnb2-141* is largely restricted to the dorsal aorta and Efnb2-159 is restricted to the intersegmental arteries, while *Efnb2-333* and *Efnb2-112* enhancers are more widely

active; there are two *Unc5b* arterial enhancers, of which *Unc5b+39* is restricted to the intersegmental arteries while *Unc5b+30* is expressed more widely. It is therefore possible that the upstream signals involved at different stages of arterial differentiation may, to some extent, target separate enhancers. Although this article has focused on transcriptional signatures of arterial versus non-arterial-specific enhancers, future research into the transcriptional differences seen between these differentially expressed arterial enhancers may therefore bring further insights into arterial transcription factors, and the manner in which upstream pathways combine to enact subtle but essential changes in gene expression.

Alongside a deficit of characterized enhancers, our understanding of vascular transcription is also affected by the considerable redundancy shown by many endothelial transcription factors. In particular, this can complicate analysis of gene disruption in animal models. A good example of this problem is the SOXF factors. SOX7, SOX17, and SOX18 not only show distinct yet overlapping expression patterns, but their ability to functionally compensate for each other can vary on different mouse backgrounds. For example, the phenotype in mice lacking SOX18 varies from essentially normal to complete loss of lymphatic ECs, with lethality depending on the mouse strain and associated variation in the ability of SOX7 and SOX17 to compensate (*Hosking et al., 2009*). SOX17 is the SOXF factor most strongly expressed in arterial ECs, and arterial defects occur after its deletion (*Corada et al., 2013*). This has resulted in the hypothesis that SOX17 selectively activates arterial genes, but this is not well supported by the results here. An alternative explanation could be that SOXF factors are required for endothelial gene expression more generally, potentially as master regulators. This aligns with their robust and primarily endothelial-specific expression (particularly after mid-gestation; *Payne et al., 2024*), and by the widespread presence of SOXF motifs and binding at endothelial enhancers of all varieties. In this second model, the loss of SOX17 may affect the arterial compartment more severely simply because it comprises the majority of arterial SOXF (so the total amount of SOXF factors is more significantly depleted in arteries than elsewhere when SOX17 is deleted), with similar explanations for the consequences of SOX7 depletion on vasculogenesis (*Lilly et al., 2017*) and SOX18 depletion on lymphangiogenesis (*Hosking et al., 2009*; *François et al., 2008*). Alternatively, *Stewen et al., 2024* have recently shown that combined depletion of SOXF factors in cultured ECs significantly reduced *Efnb2* expression while increasing *Ephb4* expression, and instead argue for a more specific role for SOXF factors in arteriovenous differentiation related to elevated SOXF levels in arterial ECs (*Stewen et al., 2024*). While this alone cannot explain the widespread binding of SOXF factors to pan-EC genes, SOXF factors are also crucial during vasculogenesis/early angiogenesis. Therefore, the role of SOXF factors in general, and SOX17 in particular, may instead be to drive a less specific angiogenic/arterial gene expression program. Supporting this, neither the *Ephb4-2* or *Mef2cF7* vein enhancers had SOXF motifs, while the role of SOXF in *CoupTFII-965* regulation may be simply related to its expression in lymphatic ECs. The paucity of defined venous/capillary enhancers currently limits our ability to make conclusions here. However, endothelial SOXF factors are clearly strongly expressed widely in the endothelium at timepoints where a much more limited number of ECs become committed to an arterial fate (perfectly illustrated in the coronary vasculature), suggesting that additional transcriptional regulators must be involved alongside SOXF to enable this exquisitely specific pattern of gene activation. While RBPJ, MEF2, and FOX factors represent obvious potential partners, none would fully explain the specificity of all arterial enhancers and they all have wider roles in the vasculature.

Complicating analysis of our arterial enhancer cohort is the possibility that all arterial enhancers are not necessarily directly activated by the same regulator(s). Instead, a transcriptional cascade may be started by the activation of just one or two early genes, which then create a more permissive environment (e.g. high concentration or post-translation modification of transcription factors) for later arterial gene expression downstream of more widely expressed transcription factors. This would align with the observed elevated levels of SOXF expression as ECs switch to an arterial fate (*Stewen et al., 2024*) and may suggest that the pathways upstream of SOXF expression play the most important role in arterial gene expression. However, a far more systematic analysis of all three SOXF loci, and the enhancers within, is required to test this hypothesis.

The relatively simple and cost-effective approach of in silico identification and zebrafish transgenesis of arterial enhancers used here had a success rate around 50%. This could doubtless be further refined (e.g. by including assessments of ERG binding, obtaining enhancer marks from in vivo

arterial cells), and made more efficient by limiting verification to F0 transgenic fish or utilizing a higher throughput assay (e.g.*Xiao et al., 2024*). However, a potentially more pressing issue is how to better understand the exact transcriptional regulators of these enhancers, a challenge shared with the gene regulatory field more widely. Transcription factors do not always bind DNA at their consensus motifs, with optimal syntax (order, orientation, and spacing of motifs) often able to compensate for poor binding sites. Additionally, the presence of multiple motifs within a single enhancer, and the ability of many transcription factors to both directly and indirectly bind enhancers, means that enhancer sequence mutational analysis can be very complicated (e.g. *Sacilotto et al., 2013*), whilst restricting this analysis to a single timepoint (usually required to make such an approach practical) can be an issue where angiogenic and arterial programmes overlap. While assessments of direct binding by ChIP-seq or similar approaches can bypass a requirement to understand the exact motifs at an enhancer, neither cultured HUVECs nor iPSC-derived arterial cells recapitulate conditions in vivo, particularly regarding availability of ligands, exposure to shear stress, and other environmental stimuli. Here, for example, the low expression of some of our arterial genes in culture HUVECs and HUAECs has probably affected verification of motifs with ChIP-seq data. Consequently, while this analysis has provided clarity as to some transcription factors involved (and not involved) in arterial gene expression, none of our observations fully explain the shared ability of these short sequences of DNA to direct arterial patterns of expression even when removed from native chromatin context and endogenous promoters. Some of these answers can be expected to come from increasing identification of new or unappreciated transcription factors specifically expressed or specifically modified in either arterial or non-arterial ECs (e.g. MECOM), better appreciation of the consensus motifs and binding patterns of proteins already known to be involved (e.g. DACH1), and improved proteomic techniques. Additionally, new iPSC models of endothelial differentiation offer the opportunity to more easily study the consequences of transcription factor perturbation during angiogenesis and arterial differentiation, and artificial intelligence, improved bioinformatic pathways, and machine learning all offer new avenues for research. It is anticipated that the cohort of in vivo-verified arterial enhancers characterized here will provide a vital platform for these future studies.

# Materials and methods

## Key resources table

| Reagent type (species) or resource | Designation | Source or reference | Identifiers | Additional information |
|---|---|---|---|---|
| Genetic reagent (*Danio rerio*) | tg(kdrl:Has.HRAS-mcherry)*s896* | *Chi et al., 2008* | ZFIN:s896 | |
| Genetic reagent (*D. rerio*) | tg(fli1:EGFP) | *Lawson et al., 2001* | ZFIN:y1 | |
| Genetic reagent (*D. rerio*) | tg(Cxcr4-194:EGFP) | This paper | ZFIN:lcr6 | Enhancer mm10 chr1:128,785,499–128,786,173 |
| Genetic reagent (*D. rerio*) | tg(Cxcr4+135:EGFP) | This paper | ZFIN:lcr7 | Enhancer mm10 chr1:128,456,948–128,457,375 |
| Genetic reagent (*D. rerio*) | tg(Cxcr4+151:EGFP) | This paper | ZFIN:lcr8 | Enhancer mm10 chr1:128,440,589–128,441,003 |
| Genetic reagent (*D. rerio*) | tg(Efnb2-333:EGFP) | This paper | ZFIN:lcr9 | Enhancer mm10 chr8:8,994,329–8,995,063 |
| Genetic reagent (*D. rerio*) | tg(Efnb2-159:EGFP) | This paper | ZFIN:lcr10 | Enhancer mm10 chr8:8,819,219–8,819,856 |
| Genetic reagent (*D. rerio*) | tg(Efnb2-141:EGFP) | This paper | ZFIN:lcr11 | Enhancer mm10 chr8:8,801,433–8,802,174 |
| Genetic reagent (*D. rerio*) | tg(Efnb2-112:EGFP) | This paper | ZFIN:lcr12 | Enhancer mm10 chr8:8,772,171–8,772,912 |
| Genetic reagent (*D. rerio*) | tg(Gja4-50:EGFP) | This paper | ZFIN:lcr13 | Enhancer mm10 chr4:127,263,607–127,264,323 |

*Continued on next page*

*Continued*

| Reagent type (species) or resource | Designation | Source or reference | Identifiers | Additional information |
|---|---|---|---|---|
| Genetic reagent (*D. rerio*) | tg(Unc5b+30:EGFP) | This paper | ZFIN:lcr14 | Enhancer mm10 chr10:60,800,677–60,801,144 |
| Genetic reagent (*D. rerio*) | tg(Unc5b+39:EGFP) | This paper | ZFIN:lcr15 | Enhancer mm10 chr10:60,792,705–60,793,377 |
| Genetic reagent (*D. rerio*) | tg(Acvrl1+6:EGFP) | This paper | ZFIN:lcr16 | Enhancer mm10 chr15:101,134,018–101,134,405 |
| Genetic reagent (*D. rerio*) | tg(Cxcl12+269:EGFP) | This paper | ZFIN:lcr17 | Enhancer mm10 chr6:117,437,567–117,438,123 |
| Genetic reagent (*D. rerio*) | tg(Gja5-78:EGFP) | This paper | ZFIN:lcr18 | Enhancer mm10 chr3:96,953,659–96,954,322 |
| Genetic reagent (*D. rerio*) | tg(Gja5-7:EGFP) | This paper | ZFIN:lcr19 | Enhancer mm10 chr3:97,025,305–97,025,791 |
| Genetic reagent (*D. rerio*) | tg(Nrp1+78:EGFP) | This paper | ZFIN:lcr20 | Enhancer mm10 chr8:128,437,292–128,437,815 |
| Genetic reagent (*Mus musculus*) | tg(Dll4in3:lacZ) | *Sacilotto et al., 2013* | Tg(Rr393-lacZ)#Sav | |
| Recombinant DNA reagent | E1b-GFP-Tol2-Gateway | Ahituv; *Birnbaum et al., 2012* | AddGene_#37846 | |
| Recombinant DNA reagent | Hsp68-LacZ-Gateway | Ahituv; *Pennacchio et al., 2006* | AddGene_#37843 | |
| Cell line (*Homo sapiens*) | HUVECs | Lonza | C2519A | Grown in EBM-2 basal medium (Lonza CC-3156) with EBM-2 SingleQuot (Lonza CC-4176) |
| Antibody | IgG control (rabbit monoclonal) | Cell Signaling Technology | CST 66362 | 1:20 (antibody total amount 0.5 µg) |
| Antibody | Sox7 (goat polyclonal) | R&D Systems | AF2766 | 1:50 (antibody total amount 0.4 µg) |
| Antibody | Sox17 (goat polyclonal) | R&D Systems | AF1924 | 1:40 (antibody total amount 0.5 µg) |
| Antibody | Sox18 (mouse monoclonal) | Abnova | H00054345-M05 | 1:100 (antibody total amount 1 µg) |
| Commercial assay or kit | NEBNext(R) Ultra II DNA Library Prep Kit | New England Biolabs | NEB E7645L | Using NEBNext Dual Index Multiplex Oligos NEB E7600S |
| Commercial assay or kit | ChIP DNA Clean & Concentrator kit | Zymo Research | D5205 | |
| Commercial assay or kit | Gateway LR Clonase II Enzyme mix | Thermo Fisher Scientific | 11791100 | |
| Commercial assay or kit | CUT&RUN Kit | Cell Signaling Technologies | CST 86652 | |
| Commercial assay or kit | pCR8/GW/TOPO TA Cloning Kit | Thermo Fisher Scientific | K250020 | |
| Software, algorithm | ZEN 2.3 lite (blue edition) | Zeiss | RRID:SCR_023747 | |
| Software, algorithm | nf-core/cutandrun | *Cheshire et al., 2023* | | Version 3.1.0 |
| Software, algorithm | Homer | *Heinz et al., 2010* | RRID:SCR_023747 | |
| Software, algorithm | vennRanges | https://rdrr.io/github/antonio-mora/vennRanges/#vignettes; *Antonio, 2019* | | Version 0.1 |
| Software, algorithm | FIJI | *Schindelin et al., 2012* | RRID:SCR_002285 | |

## Animals

All animal procedures were approved by a local ethical review committee at Oxford University and licensed by the UK Home Office and follow ARRIVE guidelines. All zebrafish were maintained in groups. F0 mosaic transgenic zebrafish embryos were generated using Tol2-mediated integration (*Kawakami, 2005*). The F1 stable transgenic lines were generated from an initial outcross of adult F0 carriers, in each case selecting founder transgenic zebrafish representative of the general expression patterns seen in F0 analysis. An intercross of adult F1 lines produced F2 lines. To enable visualization of the entire vasculature, the adult F1 transgenic lines were intercrossed with the *tg(kdrl:HRAS-mCherry) zebrafish line*. Embryos were maintained in E3 medium (5 mM NaCl; 0.17 mM KCl; 0.33 mM CaCl$_2$; 0.33 mM MgSO$_4$) at 28.5°C. Some of the embryos were incubated at 30–32°C to modify the speed of embryonic development. Some embryos were also treated with 0.003% phenylthiourea (Merck, P7629) at 24 hpf onwards, to inhibit pigmentation. To image, all embryos were dechorionated and anaesthetized with 0.01% Tricaine methanesulfonate in E3 medium. For analysis of transgenic zebrafish, single embryos were transferred into a flat-bottom 96-well plate, mounted in 0.1% TopVision low-melting point agarose (Thermo Fisher Scientific, R0801) in E3 medium with tricaine methanesulfonate (Merck, A5040). GFP and mCherry reporter gene expression was screened with a Zeiss LSM 980 (Carl Zeiss) confocal microscope at 32–72 hpf. Whole zebrafish were imaged using the 'tile scan' command, combined with Z-stack collection, at 488 nm excitation and 510 nm emission for EGFP, and at 561 nm excitation and 610 nm emission for mCherry. The eyes of the zebrafish were imaged similarly, but without tile scanning.

Adult fin analysis was performed by treating adult (3–14 months old) zebrafish with 5 g/l tricaine methanesulfonate and amputating the caudal fins with a razor blade Fins were transferred to a flat-bottom glass plate and mounted in 0.1% TopVision low-melting point agarose. GFP expression was imaged with a Zeiss LSM 980 confocal microscope as above.

E14.5 F0 transgenic mouse embryos were generated, dissected, and stained in X-gal by Cyagen Biosciences. Yolk sac was collected separately and used for genotyping. All embryos were imaged using a Leica M165C stereo microscope equipped with a ProGres CF Scan camera and CapturePro software (Jenoptik). For each enhancer, embryos were also sectioned for histological analysis to investigate X-gal staining patterns. For histological analysis, embryos were dehydrated through a series of ethanol washes, cleared by xylene, and paraffin wax-embedded. 5 or 6 µm sections were prepared, dewaxed, and counterstained with nuclear fast red (Electron Microscopy Sciences).

## Cloning

All enhancer sequences were generated as custom-made, double-stranded linear DNA fragments (GeneArt Strings, Life Technologies). The sequences of all tested enhancers are provided in Appendix 1. DNA fragments containing the enhancer sequences were cloned into the pCR8 vector using the pCR8/GW/TOPO TA Cloning Kit (Thermo Fisher Scientific, K250020) following the manufacturer's instructions. Once cloning was confirmed, the enhancer sequence was transferred from the pCR8/GW/enhancer entry vector to a suitable destination vector using Gateway LR Clonase II Enzyme mix (Thermo Fisher Scientific, 11791100) following the manufacturer's instructions. For zebrafish transgenesis, the enhancer was cloned into the E1b-GFP-Tol2-Gateway vector (*Birnbaum et al., 2012*). For mouse transgenesis, the enhancer was cloned into the hsp68-LacZ-Gateway vector.

## Enhancer mark assays

ATAC-seq in primary MAECs (SRX7016284-6) came from *Engelbrecht et al., 2020*, ATAC-seq in mouse postnatal day 6 (P6) retina ECs (MRECs) (SRX7267172-4) came from *Yanagida et al., 2020*, EP300 binding in Tie2Cre+ve cells from embryonic day (E)11.5 mouse embryos (SRX2246376-8) came from *Zhou et al., 2017*, H3K27Ac and H3K4Me1 in HUVECs, and DNAseI hypersensitivity in HUVECs and dermal-derived neonatal and adult blood microvascular ECs (HMVEC-dBl-neo/ad) came from the UCSC Genome Browser (*Rosenbloom et al., 2013*). ATAC-seq (SRX2355049 GSM2394391) and H3K27Ac (SRX2355060 GSM2394402) in cultured HAECs came from *Hogan et al., 2017*, ATAC-seq in telo-HAEC (SRX1689050 GSM6431161) came from *Schnitzler et al., 2024*, and H3K27Ac and p300 ChIP-seq in HUAECs (GSM3673407 and GSM3673413) came from *Sissaoui et al., 2020*.

## HOMER analysis on arterial enhancers

Analysis of overrepresented motifs within our validated arterial enhancer cohort was performed with HOMER's findMotifsGenome tool using the full validated region of the arterial enhancers. The analysis used the hg38 masked genome and otherwise default settings for all other parameters including randomly selected background regions.

## Transcription factor binding assays

With the exception of SOX7, SOX17 and SOX18, all transcription factor binding data was previously published, and was assessed in IGV (*Thorvaldsdóttir et al., 2013*) either through downloading from GEO or via ChIP Atlas (*Oki et al., 2018*). In every case, we first verified the correct data was accessed by reproducing images at loci used in the primary publication, and highly recommend this practice for others as errors inevitably occur during data deposition. ERG and NR2F2 binding data in HUVECs (GSM3673462 and GSM3673452) came from *Sissaoui et al., 2020*, ETS1 binding data in HUVECs after 12 hr of VEGFA stimulation (GSM2442778 SRX2452430) came from *Chen et al., 2017*. FOXO1 binding data in HUVECs (GSM3681485/6 SRX5548892) came from *Andrade et al., 2021* and FOXO1 binding in adult untreated mouse hearts (GSM4278011 SRX7586623) came from *Pfleger et al., 2020*. RBPJ binding data in HUVECs after 12 hr of VEGFA stimulation (GSM2947456 SRX3599311) came from *Wang et al., 2019*. SMAD1/5 binding in HUVECs after BMP9 stimulation (GSM684747 SRX045541) was from *Morikawa et al., 2011*, SMAD2 in HUVECs after TGFβ stimulation (GSM3955796 SRX6476491) came from *Chen et al., 2019*. MEF2C binding data in HUVECs (GSM809016 SRX100256) came from *Maejima et al., 2014*, and MEF2A binding data in mouse adult hearts (GSM3518665 SRX5146756) came from *Akerberg et al., 2019*.

For SOX7, SOX17, and SOX18 CUT&RUN, HUVECs were cultured in EBM-2 basal medium (Lonza CC-3156) supplemented with EBM-2 SingleQuot supplement and growth factor kit (Lonza CC-4176). DNA binding assays were performed using the CUT&RUN Kit from Cell Signaling Technologies (CST 86652) following the manufacturer's protocol with slight modifications. For SOX7 and SOX17, harvested cells were lightly crosslinked with 0.1% formaldehyde for 2 min and processed with buffers supplemented with 1% Triton X-100 and 0.05% SDS. SOX18 CUT&RUN was performed without crosslinking and with buffers as per standard protocol. Cells were bound to Concanavalin A beads and incubated overnight with antibodies against IgG control (CST 66362), SOX7 (R&D Systems, AF2766), or SOX17 (R&D Systems, AF1924) or SOX18 (Abnova, H00054345-M05) in wash buffer containing 0.05% digitonin. DNA around binding sites was cleaved with pAG-MNase enzyme. For SOX7 and SOX17, the released DNA was reverse-crosslinked with proteinase K and 0.1% SDS overnight at 65°C. DNA fragments were purified with a ChIP DNA Clean & Concentrator kit (Zymo Research D5205). DNA was converted into Illumina-compatible libraries with the NEBNext(R) Ultra(TM) II DNA Library Prep Kit (NEB E7645L) following the protocol described by *Liu, 2019* and using NEBNext Dual Index Multiplex Oligos (NEB E7600S). Libraries were sequenced on a NextSeq2000 (SOX17) or a NovaSeq (SOX7 and SOX18) using paired end reads. Data was processed using the nf-core/cutandrun v3.1 pipeline (10.5281/zenodo.5653535; *Ewels et al., 2020*) with the following adjustments to the default settings: `--normalisation_mode` CPM and `--trim_nextseq` 20. The CUT&RUN hg38 blacklist (*Nordin et al., 2023*) or hg19 ENCODE blacklist (*Amemiya et al., 2019*) was used during sequence alignment. Peak calling was performed with SEACR (*Meers et al., 2019*) using stringent settings, and by HOMER (*Heinz et al., 2010*) using default settings for transcription factors (-style factor). Motif analysis was performed with HOMER using 200 nt regions around peak centres. Overlap of SOX7, SOX17, and SOX18 hg19-aligned peaks with published mCherry-SOX7 data (*Overman et al., 2017*), HUVEC enhancer marks and TSS (*Sissaoui et al., 2020*) was executed using the vennRanges R package. Data has been deposited to GEO under the accession number GSE283369.

## Reanalysis of scRNA-seq data

Publicly available E12 and E17.5 scRNA-seq data from EC isolated from *BmxCreERT2;Rosa*<sup>tdTomato</sup> lineage traced murine hearts (*D'Amato et al., 2022*) was obtained from GEO (GSE214942) prior to processing FASTQ files with the 10X Genomics CellRanger pipeline (V7.0.0). RNA-seq reads were aligned to the mm10 genome reference downloaded from 10X Genomics with the addition of the T*dTomato-WPRE* sequence. Exclusion of low-quality cells with either a UMI count >100,000,, total gene count <1500, or a high proportion of reads originating from mitochondrial genes (>10%) was

performed using Scater (*McCarthy et al., 2017*). Data normalization was performed using the Multi-BatchNormalisation method prior to merging of *TdTomato*-positive and -negative datasets from individual timepoints. The top 2000 most highly variable genes (excluding mitochondrial and ribosomal genes) in the merged datasets were identified using the Seurat FindVariableFeatures method and utilized to calculate principal component analysis. Normalized data was scaled using the ScaleData function. Cell clustering was performed using the standard unsupervised graph-based clustering method implemented within Seurat (V4) (*Hao et al., 2024*). Clusters were visualized in two dimensions using UMAP based non-linear dimensional reduction following the standard Seurat (V4) workflow (*Chen et al., 2019*). Identified clusters were assigned identities based on marker genes shown to be differentially expressed between populations previously identified in the original study (*Wang et al., 2019*). Key markers include *Npr3* (endocardial), *Fabp4* (coronary vascular endothelial), and *Nfatc1* (valvular endothelial). The E12.5 sinus venosus EC cluster was assigned based in *Aplnr* as previously described (*D'Amato et al., 2022*). Arterial and venous EC clusters in the E17.5 datasets were annotated based on their enriched expression of *Gja5* and *Nr2f2*, respectively.

## Materials availability statement

The newly created zebrafish lines *tg(Cxcr4-194:EGFP)*, *tg(Cxcr4+135:EGFP)*, *tg(Cxcr4+151:EGFP)*, *tg(Efnb2-333:EGFP)*, *tg(Efnb2-159:EGFP)*, *tg(Efnb2-141:EGFP)*, *tg(Efnb2-112:EGFP)*, *tg(Gja4-50:EGFP)*, *tg(Unc5b+30:EGFP)*, *tg(Unc5b+39:EGFP)*, *tg(Acvrl1+6:EGFP)*, *tg(Cxcl12+269:EGFP)*, *tg(Gja5-78:EGFP)*, *tg(Gja5-7:EGFP)* and *tg(Nrp1+78:EGFP)*, and the plasmids used to generate them are all available on request from the corresponding author. The SOX7, SOX17, and SOX18 CUT&RUN data is deposited at GEO under the accession number GSE283369.

## Acknowledgements

We thank Nadav Ahituv for providing the GW vectors. This work was supported by the BHF (FS/1735/32929 and FS/SBSRF/22/31037 to SDV and SN; FS/IPBSRF/23/27085 to IRM), the Oxford BHF Centre of Research Excellence (RE/18/3/34214), the Fondation Leducq, and Ludwig Cancer Research Ltd.

## Additional information

### Funding

| Funder | Grant reference number | Author |
| --- | --- | --- |
| British Heart Foundation | FS/1735/32929 | Svanhild Nornes<br>Sarah De Val |
| British Heart Foundation | FS/SBSRF/22/31037 | Svanhild Nornes<br>Sarah De Val |
| British Heart Foundation | FS/IPBSRF/23/27085 | Ian R McCracken |
| British Heart Foundation | RE/18/3/34214 | Sarah De Val |
| Fondation Leducq | 18CVD03 | Susann Bruche |
| Ludwig Institute for Cancer Research | | Sarah De Val |

The funders had no role in study design, data collection and interpretation, or the decision to submit the work for publication.

### Author contributions

Svanhild Nornes, Supervision, Investigation, Methodology, Writing – review and editing; Susann Bruche, Formal analysis, Investigation, Writing – review and editing; Niharika Adak, Investigation; Ian R McCracken, Data curation; Sarah De Val, Conceptualization, Resources, Data curation, Formal analysis, Supervision, Funding acquisition, Investigation, Methodology, Writing – original draft, Project administration, Writing – review and editing

### Author ORCIDs

Svanhild Nornes (ID) https://orcid.org/0000-0002-5301-5252
Susann Bruche (ID) https://orcid.org/0000-0002-5814-7166
Sarah De Val (ID) https://orcid.org/0000-0002-2566-2348

### Ethics

All animal procedures were approved by a local ethical review committee at Oxford University and licensed by the UK Home Office, license number PP1224162.

### Decision letter and Author response

Decision letter https://doi.org/10.7554/eLife.102440.sa1
Author response https://doi.org/10.7554/eLife.102440.sa2

## Additional files

### Supplementary files

MDAR checklist

### Data availability

Cut&Run data for Sox7, Sox17 and Sox 18 have been deposited at GEO under the accession number GSE283369.

The following dataset was generated:

| Author(s) | Year | Dataset title | Dataset URL | Database and Identifier |
|---|---|---|---|---|
| Nornes S, Bruche S, Adak N, McCracken I, De Val S | 2024 | SOX7, SOX17 and SOX18 DNA binding in baseline HUVEC | https://www.ncbi.nlm.nih.gov/geo/query/acc.cgi?acc=GSE283369 | NCBI Gene Expression Omnibus, GSE283369 |

The following previously published datasets were used:

| Author(s) | Year | Dataset title | Dataset URL | Database and Identifier |
|---|---|---|---|---|
| Engelbrecht E, Levesque MV, He L, Vanlandewijck M, Betsholtz C, Hla T | 2019 | Sphingosine 1-phosphate-regulated transcriptomes in heterogenous arterial and lymphatic endothelium of the aorta | https://www.ncbi.nlm.nih.gov/geo/query/acc.cgi?acc=GSE139065 | NCBI Gene Expression Omnibus, GSE139065 |
| Engelbrecht E, Yanagida K, Hla T | 2019 | Sphingosine 1-phosphate receptor signaling establishes AP-1 transcriptional factor gradients and permits retinal endothelial specialization | https://www.ncbi.nlm.nih.gov/geo/query/acc.cgi?acc=GSE141440 | NCBI Gene Expression Omnibus, GSE141440 |
| Zhou P, Gu F, Zhang L, Akerberg BN, Ma Q, Li K, He A, Lin Z, Stevens SM, Zhou B, Wt PU | 2017 | Mapping cell type-specific transcriptional enhancers using high affinity, lineage-specific p300 bioChIP-seq | https://www.ncbi.nlm.nih.gov/geo/query/acc.cgi?acc=GSE88789 | NCBI Gene Expression Omnibus, GSE88789 |
| Romanoski CE, Hogan NT | 2017 | Genome-wide map of HAEC chromatin landscape under resting and TNFa, IL1b, and OxPAPC stimulation, with corresponding transcription factor binding and RNA expression | https://www.ncbi.nlm.nih.gov/geo/query/acc.cgi?acc=GSE89970 | NCBI Gene Expression Omnibus, GSE89970 |

*Continued on next page*

*Continued*

| Author(s) | Year | Dataset title | Dataset URL | Database and Identifier |
|---|---|---|---|---|
| Schnitzler GR, Kang H, Lee-Kim V, Xr MA, Zeng T, Vellarikkal SK, Zhou R, Guo K, Sias-Garcia O, Bloemendal A, Munson G, Guckelberger P, Nguyen TH, Bergman DT, Atri D, Cheng N, Cleary B, Lander ES, Finucane HK, Gupta RM, Engreitz JM | 2022 | High-content CRISPR screens link coronary artery disease genes to endothelial cell programs [ATACseq] | https://www.ncbi.nlm.nih.gov/geo/query/acc.cgi?acc=GSE210489 | NCBI Gene Expression Omnibus, GSE210489 |
| Lawson N | 2020 | Identification and characterization of artery and vein enhancers in the human genome | https://www.ncbi.nlm.nih.gov/geo/query/acc.cgi?acc=GSE128382 | NCBI Gene Expression Omnibus, GSE128382 |
| Huan C, Day DS, Fu Y, Sun Y, Wang S, Zhang F, Yan P, Gu F, Stevens SM, Seidman JG, Han Z, Park PJ, Zhang B, Wt PU | 2016 | VEGF promotes RNAPII pausing release through ETS1 to stimulate angiogenesis | https://www.ncbi.nlm.nih.gov/geo/query/acc.cgi?acc=GSE93030 | NCBI Gene Expression Omnibus, GSE93030 |
| Andrade J, Zimmermann B, Grosso AR, Potente M | 2021 | Genome-wide maps of FOXO1 binding sites in human endothelial cells | https://www.ncbi.nlm.nih.gov/geo/query/acc.cgi?acc=GSE128635 | NCBI Gene Expression Omnibus, GSE128635 |
| Pfleger J, Koch WJ | 2020 | FoxO1 Binding in Cardiac Hypertrophy | https://www.ncbi.nlm.nih.gov/geo/query/acc.cgi?acc=GSE144011 | NCBI Gene Expression Omnibus, GSE144011 |
| Wang S, Chen J, Garcia S, Liang X, Zhang F, Fu Y, Yan P, Yu H, Wei W, wang J, Le H, Han Z, Day DS, Stevens SM, Zhang Y, Park PJ, Sun K, Yuan G, Wt PU, Zhang B | 2018 | Dynamic and integrated transcriptional code orchestrates the angiogenic response [Seq] | https://www.ncbi.nlm.nih.gov/geo/query/acc.cgi?acc=GSE109625 | NCBI Gene Expression Omnibus, GSE109625 |
| Morikawa M, Koinuma D, Tsutsumi S, Vasilaki E, Heldin C, Aburatani H, Miyazono K | 2011 | SMAD1/5 binding regions of human umbilical vein endothelial cells (HUVECs) treated with BMP | https://www.ncbi.nlm.nih.gov/geo/query/acc.cgi?acc=GSE27634 | NCBI Gene Expression Omnibus, GSE27634 |
| Simons M, Chen P | 2019 | Endothelial TGFb signaling drives vascular inflammation and atherosclerosis [ChIP-Seq] | https://www.ncbi.nlm.nih.gov/geo/query/acc.cgi?acc=GSE134556 | NCBI Gene Expression Omnibus, GSE134556 |
| Maejima T, Kohro T, Tsutsumi S, Yamamoto S, Kimura H, Kodama T, Wada Y | 2013 | Genome-wide maps of MEF2C and H3K27ac localization in HUVECs | https://www.ncbi.nlm.nih.gov/geo/query/acc.cgi?acc=GSE32644 | NCBI Gene Expression Omnibus, GSE32644 |
| Akerberg BN, Gu F, Wt Pu | 2019 | A reference map of cardiac transcription factor chromatin occupancy identifies dynamic and conserved transcriptional enhancers | https://www.ncbi.nlm.nih.gov/geo/query/acc.cgi?acc=GSE124008 | NCBI Gene Expression Omnibus, GSE124008 |

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
