## [Editor Report]

This work represents a significant milestone in understanding the regulatory logic underlying arterial identity. It offers an exceptional repository of validated arterial enhancers that drive expression across different vascular compartments. Moreover, it identifies the upstream transcription factor binding sites within these arterial enhancers. As such, this study will be of broad interest to developmental biologists, vascular researchers, and cell biologists.

---

## [Decision Letter]

[Editors' note: this paper was reviewed by Review Commons.]

---

## [Author Response]

1. General Statements

We thank all three reviewers for their overwhelmingly positive reviews. As they noted, this paper not only advances our knowledge of the pathways regulating arterial differentiation, but also provides a valuable and easy-to-access resource that will be appreciated by a wide range of researchers. This obviously includes those interested in arterial differentiation, vascular development and associated diseases. However, we also anticipate this resource will be used more widely, for example providing an interesting and well validated enhancer cohort resource for epigenetic researchers interested in enhancer action and specificity more generally, and as a useful training set for testing bioinformatic/AI pipelines.

While resource papers more often focus on generating large novel datasets or analysis pipelines, this can leave a gap between the existence of data and the ability of researchers to apply it to their ongoing studies. This problem is clearly evident in the vascular field, where it is common to see otherwise strong research papers rely on outdated analysis of promoter/intronic regions to link their upstream pathways to transcriptional activation. Additionally, conclusions from complex pan-genomic analyses are often made without ever verifying any “enhancer” regions in transgenic models, influenced by prohibitory costs and perceived complexity of such analysis. As well as the potential for errors and incorrect assumptions, a reliance on poorly defined enhancer/promoter regions prevents further researchers building on these initial observations, to the general detriment to the field. Here, we have not only provided a detailed and standardized characterization of all known vascular enhancers but also an accessible enhancer analysis blueprint for others to follow. This permits all researchers easy access the sequence and transcriptional profile for every enhancer/gene/transcription factor investigated here, whilst also enabling them to follow a similar pathway to identify, annotate and characterize novel enhancers for any gene of interest. We note that the bioRxiv version of our paper has already been cited by Coronado et al., medRxiv https://doi.org/10.1101/ 2023.10.27.23297507, where it enabled them to easily link a GWAS association at a non-coding region near the chemokine CXCL12 with a validated arterial enhancer (Cxcl12-269). No doubt the enhancer annotation provided in our paper will also enable these researchers to better examine the various linkage SNPs overlapping Cxcl12-269 to more directly link phenotype to transcriptional motif changes.

We would like to conclude with three comments from our reviewers which emphasizes the strength of this paper. Reviewer 1: “This work represents a significant milestone in the systematic understanding of how arterial gene expression is regulated. Overall, this study offers a powerful resource for understanding arterial gene regulation and conducting genome-wide studies of arterial enhancers”. Reviewer 2: “This very well-done study…advances our knowledge on the field of vascular biology as it not only proposes potential enhancers but also goes on to validation of the enhancers”. Reviewer 3: “This novel work establishes an important foundation for future understanding of how TFs may interact to determine arterial specification”.

This Revision Plan covers all issues/comments made by the reviewers, although some of the text has been abbreviated. We have also separately submitted a non-abbreviated point-by-point response to reviewers.

2. Description of the planned revisionsReviewer 1: The co-localization of the enhancer expression in the endothelium was done using endothelial marks expressed in both venous and arterial EC (kdrl). To fully distinguish if the expression is venous or arterial endothelial compartment colocalization with Tg expressed in arterial (flt1) or venous (lyve1) EC would be informative. In addition, it is striking that cxc4+135 drives the expression in nearly every ISV as cxcl12+269 only every other. Similarly, not all the enhancers are enriched in the DA to the same level. Is there biological significance to this? could authors discuss these results further? The pattern of expression of the unc5b-identified enhancer is also striking, does this reflect the known roles of unc5b in the vascular formation?

Planned Revision 1. Updated Figure 3 and new Table to better describe and characterize the expression of the arterial enhancers in transgenic zebrafish.

We agree with the reviewer that a more detailed description of arterial-venous specificity of each enhancer could be included. The diversity of enhancer expression patterns within the arterial compartment is notable (and really very interesting) and could be discussed in greater depth.

In the original manuscript, the expression pattern of each enhancer within the vasculature was primarily assessed at 2 days post fertilization (2dpf) in Figure 1-2. This identified arteries using direction of blood flow and known anatomical information, as arterial development in 2pdf zebrafish is very stereotypical and already well characterized. The original Figure 3A includes a more detailed assessment of arterial-venous specificity at 3dpf for four arterial enhancers (Cxcr4+135, Cxcr4+151, Gja5-78 and Gja57, chosen as enhancers representing the four types of expression patterns seen). We will now extend this more detailed analysis to all arterial enhancer:GFP lines. This analysis uses kdrl-mCherry to mark the entire vasculature comparative to the expression of the arterial enhancers (GFP). This allows us to clearly identify the intersegmental arteries (as opposed to intersegmental veins) by looking for direct connection to the dorsal aorta, and by assessing the direction of blood flow within these vessels. This analysis is done at 3dpf to give time for the intersegmental arteries to acquire identity and connect definitively with the dorsal aorta, and for the diminishment of any GFP expression originating from the initial sprouting from the dorsal aorta. By extending this analysis to the other arterial enhancer zebrafish lines shown in Figure 2, we will be able to more clearly classify the activity of each enhancer within different vascular beds. Detailed expression information will also be recorded in a new Table better detailing the timing and specificity of activity of each enhancer.

We chose not to use arterial or venous “marker lines” (e.g. Flt1:reporter or Lyve1:reporter) for the simple reason that these are also enhancer:reporter transgenes, and therefore are not necessarily definitive of the arterial or venous lineage per se (e.g. Flt1:GFP expression is controlled by the transcription factors binding the Flt1 enhancer in the same way that Cxcr4+135 and the others are, with the added caveat that the transcriptional regulation of the Flt1 and Lyve enhancers are not well defined).

We felt that morphological determination based on direct connections and blood flow direction was therefore more accurate.

The extension of Figure 3A to all enhancer lines will also permit us to more clearly classify the activity of each arterial enhancer within different beds and at different time points. Currently there were no clear links between a particular transcription factor motif/binding and expression pattern, something that is discussed briefly in the original Results and Discussion sections. However, the expansion of Figure 3A to all enhancers, and the creation of a Table summarizing this more systematically will make the link (or lack of one) between expression patterns within the arterial tree and TF motifs easier to appreciate and discuss.

Reviewer 2 Major Comment 4. The analysis of the enhancers is only done during development. Is the activity of these enhancers maintained through life or only important for artery vs vein determination? Is the expression of the different enhancer reporters maintained into adulthood?

Planned Revision 2. Analysis of expression of arterial enhancers in adult transgenic zebrafish fins.

We agree this would be interesting to ascertain. To address this, we will include an examination of the activity of each arterial enhancer:GFP transgene in adult fish fins in the fully revised version of this paper. The vessels in the adult fin are accessible for analysis without the need to cross the fish into a *casper* background, which would be beyond the timescale of this project. We have already conducted a feasibility study on four arterial enhancers:GFP lines (Gja5-7:GFP, Gja5-78:GFP, Gja4+40:GFP and Efnb2-333:GFP), which found that all four were active and arterial-specific in the adult zebrafish fin.

Reviewer 3 Major Comment 3: SoxF family TFs. Among the 3 members of SoxF TFs, only Sox17 and Sox7 were assessed. Though not specific, Sox18 is highly expressed in the arteries. On the contrary, Sox7 is highly expressed in the vein and shows weak expression in arterial ECs (PMID: 26630461).

Planned Revision 3. Inclusion of SOX18 ChIP-seq data in analysis

We agree. We will conduct a new ChIP-seq/CUT&RUN analysis and include assessment of SOX18 binding in our final revised manuscript. We have identified a suitable antibody for this analysis.

Reviewer 3 Comment 4: Figure 4 e14.5 mouse embryos. If the observation aims to assess the dorsal aorta, it would be better to use mouse embryos at mid-gestation (e9.5-10.5), when the paired DAs are formed with arterial identity but haven't been remodelled and fused as one single aorta. The morphological data in this figure would be better to show the colocalization of LacZ expression and an arterial marker (e.g. Sox17) using immulfluorescence staining instead of purely lacZ.

Planned Revision 4. Include images from sections through e14.5 embryos to demonstrate expression patterns

We are unable to perform immunofluorescence in the e14.5 transgenic embryos due to the fixation and staining solutions used for X-gal staining (which was done by an external company and could not be altered), but agree additional data would better demonstrate arterial endothelial specificity. We will expand the analysis of sectioned embryos (currently restricted to just the Efnb2-333:LacZ transgene) to all enhancers shown in Figure 4. This analysis has some limitations due to infiltration of the X-gal solution to deeper tissues, but is anticipated it will clearly show enhancer activity in arterial endothelial cells rather than venous ECs or smooth muscle cells.

It is important to emphasize that this experiment was primarily conducted to demonstrate that our enhancers were arterial enriched in both zebrafish and mouse transgenesis. We feel this is clearly shown with the e14.5 transgenic embryos as originally shown. We chose e14.5 because it matched the timepoints used for the single cell transcriptomics from which we selected the target arterial identity genes, and feel it is also a good match to 2-3 dpf zebrafish in terms of arterial differentiation mechanisms. We agree that E9-10 would have also been an additional useful timepoint, but we do not have the resources to generate this data nor consider it essential for the conclusions of our work here.

3. Description of the revisions that have already been incorporated in the transferred manuscriptReviewer 1:In literature, the term 'deep conservation' refers to evolutionary conservation (genomic sequence preservation) in a wide range of species. Therefore, the additional classification presented by the authors based on the surrounding sequence is not clear. As, the KLF motifs in the Ece1in1, which is conserved between mouse and human, are defined as "deeply conserved". However, the FLK motif in the following enhancer, Flk1in10 (one line below), gets classified as non-deeply conserved, despite also being conserved between mouse and human. Thus, in the current form, there is a contradiction in the way the authors use the term 'deeply conserved' and the accepted meaning of this term. To avoid confusion, it would be important to revise this nomenclature.

Incorporated Revision 1. Change in nomenclature to alter the term “deep conservation” to “strong conservation” and new text to better describe this.

We have altered our nomenclature. This is explained in the relevant Results sections: “Because the level of conservation of motifs can often be an indication of their importance to enhancer activity, we classified each motif into three categories: strongly conserved (motif conserved to the same depth of the surrounding sequence), weakly conserved (motif conserved in orthologous human enhancer but not to the same depth as the surrounding sequence) and not conserved (motif is not conserved within the orthologous human sequence)”.

Two enhancers (Unc5b-57 and Cdh1-1) were only conserved human-mouse, therefore each TF motif within these enhancers could be annotated as both weakly and strongly conserved. As the reviewer noted, this does create confusion. We have now adjusted Figure 5 to use a distinct shape for motifs for which no distinction between weak and strong motif can be made. This does not cover Ece1in1, which is conserved human-mouse-tenrec but was erroneously originally labelled human-mouse only. This error has been corrected.

Reviewer 1 Minor comment: Figure 1 and 2 for non-zebrafish readers it would be useful to indicate in Figures 1 and 2 the non EC expression that can be observed in the embryos.

Incorporated Revision 2. Improved labeling in Figure 1 and Figure 2.

In addition to arterial expression, a number of the enhancer:GFP transgenes also showed GFP expression within the neural tube. In addition, some transient transgenic embryos also showed ectopic expression in muscle fibers. These have now been clearly labelled on the images in Figure 1 and 2.

Reviewer 2 Major Comment 2:The human data comes from vein endothelial or microvasculature endothelial cells. Specially because some of the enhancers identified by the authors drive also vein expression, could the authors discriminate whether this is due to the identification coming from vein cells. Is there available data from HAECs? Would this not be conceptually more correct that using vein endothelial cells data? This should be at least discussed in the paper.

Incorporated Revision 3. Inclusion of enhancer marks from HAECs, telo-HAECs and HAUECs

We have now included a comparison with enhancer marks from HAECs, telo-HAECs and HUAECs as a new Figure 1—figure supplement 4. The enhancer marks seen in these cells were very similar to those in the HUVEC and microvascular cells already surveyed. Had enhancer marks within HAECs/telo-HAEC/HUAECs been included as a human enhancer mark in our initial survey, it would have been unlikely to have altered our analysis, although we agree it would have made it more conceptually correct. We chose not to go back and engineer this into our original enhancer selection rational however as we felt it would be intellectually dishonest. A paragraph has been added to the Results section about this analysis:

“We also considered whether the use of vein-origin (HUVEC) and microvascular-origin (HMVEC-dBl-neo/ad) ECs in our analysis of human enhancer marks may have affected the accuracy of our putative enhancer selection by expanding our analysis to enhancer marks in arterial-origin ECs. However, analysis of enhancer marks in human aortic endothelial cells (HAEC and telo-HAECs) and human umbilical artery endothelial cells (HUAECs) showed a very similar pattern, and identified the same set of putative enhancers as when HUVEC/HMVEC data was considered (Figure 1 – supplemental figure 4). This suggests that the arteriovenous original of cultured cells did not significantly influence putative enhancers marks, further emphasizing the challenges of using selecNve enhancer marks in such lines to predict expression patterns in vivo.”

Reviewer 4 Major Comment 1.Choice of arterial genes is slightly biased. Acvrl1/Alk1 is not enriched in arterial ECs. Sema3G, which is highly expressed in arterial ECs, is missing. UNC5B is enriched in arterial ECs but also expressed by sprouting ECs (PMID: 38866944).

Incorporated Revision 4. Better explanation of choice of arterial genes

The choice of arterial genes was already discussed in the Results and Discussion section. However, we have now edited the first Results section to better explain gene selection: “It is therefore clear that a better understanding of the regulatory pathways directing arterial differentiation requires the identification and characterization of a larger number of arterial enhancers directing the expression of key arterial identity genes. To identify a cohort of such enhancers, we looked in the loci of eight non-Notch genes: Acvrl1(ALK1) Cxcr4, Cxcl12, Efnb2, Gja4(CX37), Gja5 (CX40), Nrp1 and Unc5b. Although not a definitive list of arterial identity genes, single cell transcriptomic analysis indicates these genes are all significantly enriched in arterial ECs4,20, and are commonly used to define arterial EC populations in mouse and human scRNAseq analysis^4,5,20,54^. Additionally, single-cell transcriptomic data indicates that arterial ECs can be divided into two subgroups^4,20^. The genes selected here are equally split between subgroups (Acvrl1, Cxcl12, Gja5 and Nrp1 from the mature arterial EC subgroup, Cxcr4, Efnb2, Gja4 and Unc5b from the less mature/arterial plexus/pre-arterial EC subgroup)^4,20^. We did not exclude genes also implicated in angiogenesis/expressed in sprouting ECs, as these genes formed that vast majority of those associated with the less mature EC subgroup”. Discussions about the overlap with angiogenic/sprouting genes can also be found in the original Results section (angiogenic expression of arterial genes is discussed within the *MEF2* and RBPJ sections) and in the Discussion (paragraph 2, referring to different expression patterns within arterial ECs).

When we started this project, scRNA-seq datasets in the developing endothelial vasculature were less available than now. Consequently, we initially based our choice of genes on data from Raftrey *et al.,* Circ Res 2021 (available earlier on bioRxiv), which was focused on mouse coronary arterial ECs at the timepoints that arteries differentiate. This found *Acvrl1* to be arterial enriched (not a novel observation, many publications treat *Acvrl1* as arterial specific or arterial-enriched) and did not list *Sema3g*. We also considered a wider dataset from mouse and human mid-gestation embryos when available (Hou *et al.,* Cell Research 2022). However, it is important to note that we did not aim to investigate every arterial enriched gene, rather to use these datasets to help identify loci associated with gene expression patterns which indicated a high likelihood of containing arterial enhancers active during arterial differentiation.

Sc-RNAseq data from both Raftery et al., and Hou et al., indicated that arterial ECs are subdivided into two groups, reflecting maturity but also potentially slightly different developmental trajectories. The genes studied here were therefore selected to evenly cover both subgroups, with *Acvrl1*, *Cxcl12*, *Gja5* and *Nrp1* primarily restricted to the mature arterial EC subgroup, while *Cxcr4*, *Efnb2*, *Gja4* and *Unc5b* were also expressed in the less mature/arterial plexus/pre-arterial EC subgroup. It is notable that genes within the latter subgroup are also associated with angiogenic/sprouting ECs (*Dll4* also belongs to this subgroup), which likely indicates biological links between angiogenesis and arterial identity rather than a problem in gene choice and specificity.

Incorporated Revision 5. Consideration of two additional putative enhancers Efnb2-159 and Cxc4+119*.*

This was not in response to reviewers but instead in response to an error spotted during revision. The putative arterial enhancers Efnb2-159 and Cxcr4+119 were initially omitted in error yet both regions reach the standard of testable putative enhancers (this has been noted in small changes to Figure S1 and Table S2). When tested in zebrafish transient transgenic embryos, Cxcr4+119 was inactive whilst Efnb2-159 was active in arterial endothelial cells. The relevant tables and figures have been adjusted to reflect these changes, the most significant of which are the inclusion of Efnb2-159 positive zebrafish in Figure 1 (and the necessity to create an additional supplemental Figure (S3) to accommodate the increased number of images), and analysis of Efnb2-159 transcription factor motifs/binding as part of Figure 5 and 6. This brings the total of novel robust arterial enhancers identified here to 16, and the total of validated arterial enhancers in all literature to 24. No conclusions were altered by the inclusion of this data.

Reviewer 1 Minor Comment:Details on how the corresponding non-coding regions between mice and humans were established are missing, what alignment tool was used?

Incorporated Revision 6. Altered text to address missing or incorrect pieces of information

This information has now been included in the relevant Results section: “Orthologous human enhancer sequences were identified for every enhancer using the Vertebrate Multiz Alignment & Conservation Track on the UCSC genome browser”

Reviewer 1 Minor Comment:Not sufficient details are provided for the re-analysis of siRNA data. E.g., which clustering method was used? How the clusters were assigned to cell identities?

The information detailing the re-analysis of scRNA data has now been expanded in the methods section.

Reviewer 1 Minor Comment:Details about the first HOMER analysis (in the assessment of transcription factor motifs and binding patterns at arterial enhancers) seem to be missing from the methods section.

This has now been included in the methods section.

Reviewer 1 Minor Comment:Pg 12: "For ETS, 23/23 arterial enhancers contained at least one conserved motif (all "deeply" conserved to the same depth as the surrounding enhancer, see S7)". Is it S8, where conservation is indicated?

We have corrected this error in the text – no figure actually needed to be referenced here as the previous sentence contained the full list of relevant figures to this statement (Table 2 and Figures 5 and S9, previously called S8, are the places to see this information).

Reviewer 1 Minor Comment Table S1:Please, indicate in the legend what the asterisk in the H DNAseI column stands for

The asterisk indicates where DNaseI hypersensitivity is also seen in multiple non-EC lines. This explanation has been added to the legend.

Reviewer 3 Minor inaccuracy in Intro/paragraph 3:Though sox17 is reported as indispensable for arterial specification (PMID: 24153254), losing a single SoxF factor does not seem to completely compromise the arterial program (PMID: 24153254, PMID: 26630461). A combined loss of Sox17/18, or Sox 7/17/18, seems to do the job (PMID: 26630461).

We have altered this section to “The evidence linking SOXF transcription factors to arterial differentiation is more extensive, with loss of either SOX17 (the SOXF factor most specific to arterial ECs) or SOX7 resulting in arterial defects^21–24^. Whilst losing a single SOXF factor does not entirely compromise the arterial program, arterial differentiation appears absent after compound Sox17;Sox18 and Sox7;Sox17;Sox18 deletion, although this occurs alongside significantly impaired angiogenesis and severe vascular hyperplasia^21–24^.” PMID 24153254 is reference 23, PMID 26630461 is reference 24.

4. Description of analyses that authors prefer not to carry outReviewer 1 Minor Comment Figure S8:The phrasing "conserved to animal" in Figure S8 is misleading. There is no difference in something being conserved to tenrec or manatee, as both are Afrotherians. Hence, the data show that both Efnb2-141 and Ephb4-2 were present in the common ancestor of Afrotherians and humans, namely the ancestor of all placentals. Instead, it would be good to indicate the phylogenetic group for which the presence of the enhancer can be inferred (in this case, Placentalia).

Whilst I appreciate the point, it is the exact sequence that is important here – obviously tenrec and manatee are similar species but still contain differences in nucleotide sequences. The information about conservation leads the reader to the exact species with which the comparison is being made. We tried to restrict this to just one species per phylogenetic group (e.g. tenrec, opossum, chicken, zebrafish) but occasionally this was not possible.

Reviewer 2 Major Comment 1:In their identification of enhancers, the authors consider a candidate every enhancer that has a putative mark in both mouse and human. Nevertheless, all the human data comes from in vitro analysis. Considering how much cell culture affects endothelial cell identity, inducing effects like EndoMT, would this have any effect on the enhancer selection? Would it be possible to search any human in vivo data? Would this allow for even stronger and more relevant sequences?

We agree that the use of human endothelial cells in culture raises some potential issues. However, we stress that the mouse EC enhancer marks, which played a key role in defining putative enhancers, come from in vivo analysis (E11 embryos, P6 retina and adult aorta), limiting the potential for significant impact from cell culture-induced issues. Whilst we would have enthusiastically incorporated human in vivo data had it been available, our approach was still indisputably successful at identifying arterial enriched/specific enhancers.

We consider it unlikely that culture/identity-related problems with human cultured ECs led to a significant undercount of enhancers, in part because comparatively few regions with enhancer marks in mouse in vivo ECs were excluded due to the absence of human enhancer marks. In fact, *Cxcr4*, *Cxcl12*, and *Gja5* were poorly transcribed in the human cell lines studied here and consequently only enhancer marks in mouse were used to define putative enhancers for these three genes (this is clearly stated in the Results section). If a similar rational had applied to the remaining five genes, only an additional six putative enhancers would have been tested (one for *Gja4*, two for *Nrp1* and three for *Unc5b*). However, we felt it made sense to include analysis of human enhancer marks for these five genes, as all were expressed in the human ECs used (as indicated by H3K1Me3 and DNaseI hypersensitivity at promoter regions) and orthologous human enhancers were identified for all. Additionally, our retrospective analysis of previously described mammalian in vivo-validated EC enhancers (Table S1 in the original manuscription, including eight arterial enhancers) found that all 32 were marked by at least one enhancer mark in human samples (1/32 did not contain mouse enhancer marks). We also tested eleven regions that did not reach our putative enhancer threshold, including five with only mouse marks. None of these directed expression in transgenic analysis.

Reviewer 2 Major Comment 3:Although the authors use the mouse embryo to further validate their finding beyond the zebrafish, the expression are a bit different. While on the fish the enhancers label smaller vessels of arterial identity, in the mouse, only bigger arteries are marked. Is this defined by the time of the analysis?

This experiment was conducted to demonstrate that activity of these enhancers are arterial enriched in both zebrafish and mouse, and feel this is clearly shown by the current data. Whilst I do not really agree that the expression pattern is different (for example, the Gja5 enhancers are more restricted to the major arteries in both zebrafish and mouse, compared to the more widely expressed Efnb2-333), this is challenging to ascertain at a single time-point in a transient transgenic mouse assay. Whilst it would be potentially interesting to better assess the activity of these enhancers over time in mice, this would be an expensive and lengthy experiment (multiple stable lines would need to be established and characterized for each enhancer) which would not add a notable benefit to this paper.

Reviewer 3 Major Comment 2:Exclusion of Notch genes. Although the reason for choosing non-notch genes and excluding notch genes for screening is addressed in this paper, it would be interesting to examine how the arterial enhancers identified in this study are present in the Notch genes, especially Dll4 (enriched in arterial and sprouting ECs) and Jag1 (enriched in arterial ECs).

Previous work from our lab and others has already examined arterial enhancers for many Notch pathway genes. We already included these enhancers (including two for *Dll4*, one for *Notch1* and one of *Hey1*) in all our later analysis (Figure 5-6 and relevant supplemental figures). Whilst *Jag1* is not included in this analysis, we do not feel additional enhancer identification is needed to support the conclusions of this paper.